# Net Cloud Thinning, Low-Level Cloud Diminishment, and Hadley Circulation Weakening of Precipitating Clouds with Tropical West Pacific SST Using MISR and Other Satellite and Reanalysis Data

**Terence L. Kubar [1,2,\*] and Jonathan H. Jiang [2]**

1   Joint Institute for Regional Earth System Science and Engineering, University of California Los Angeles, 4242 Young Hall, 607 Charles E. Young Drive East, Los Angeles, CA 90095-7228, USA
2   Jet Propulsion Laboratory, California Institute of Technology, 4800 Oak Grove Drive, MS 233-300, Pasadena, CA 91109, USA; Jonathan.H.Jiang@jpl.nasa.gov
\*   Correspondence: tkubar@ucla.edu; Tel.: +1-818-354-0145

**Abstract:** Daily gridded Multi-Angle Imaging Spectroradiometer (MISR) satellite data are used in conjunction with CERES, TRMM, and ERA-Interim reanalysis data to investigate horizontal and vertical high cloud structure, top-of-atmosphere (TOA) net cloud forcing and albedo, and dynamics relationships against local SST and precipitation as a function of the mean Tropical West Pacific (TWP; 120°E to 155°W; 30°S–30°N) SST. As the TWP warms, the SST mode (~29.5 °C) is constant, but the area of the mode grows, indicating increased kurtosis of SSTs and decreased SST gradients overall. This is associated with weaker low-level convergence and mid-tropospheric ascent ($\omega_{500}$) over the highest SSTs as the TWP warms, but also a broader area of weak ascent away from the deepest convection, albeit stronger when compared to when the mean TWP is cooler. These associated dynamics changes are collocated with less anvil and thick cloud cover over the highest SSTs and similar thin cold cloud fraction when the TWP is warmer, but broadly more anvil and cirrus clouds over lower local SSTs (SST < 27 °C). For all TWP SST quintiles, anvil cloud fraction, defined as clouds with tops > 9 km and TOA albedos between 0.3–0.6, is closely associated with rain rate, making it an excellent proxy for precipitation; but for a given heavier rain rate, cirrus clouds are more abundant with increasing domain-mean TWP SST. Clouds locally over SSTs between 29–30 °C have a much less negative net cloud forcing, up to 25 W m$^{-2}$ greater, when the TWP is warm versus cool. When the local rain rate increases, while the net cloud fraction with tops < 9 km decreases, mid-level clouds (4 km < $Z_{top}$ < 9 km) modestly increase. In contrast, combined low-level and mid-level clouds decrease as the domain-wide SST increases (−10% deg$^{-1}$). More cirrus clouds for heavily precipitating systems exert a stronger positive TOA effect when the TWP is warmer, and anvil clouds over a higher TWP SST are less reflective and have a weaker cooling effect. For all precipitating systems, total high cloud cover increases modestly with higher TWP SST quintiles, and anvil + cirrus clouds are more expansive, suggesting more detrainment when TWP SSTs are higher. Total-domain anvil cloud fraction scales mostly with domain-mean $\omega_{500}$, but cirrus clouds mostly increase with domain-mean SST, invoking an explanation other than circulation. The overall thinning and greater top-heaviness of clouds over the TWP with warming are possible TWP positive feedbacks not previously identified.

**Keywords:** remote sensing; tropical convection; cloud fraction; cloud radiative forcing; cloud feedbacks; precipitation

## 1. Introduction

The nature of tropical convection—including its vertical distribution, horizontal extent, and optical properties, and associated top-of-atmosphere (TOA) radiative effects—is associated with SST, SST gradients, and regional and large-scale circulation and dynamics. SST is highly relevant as ocean–atmosphere exchanges drive the planetary boundary layer temperature, which largely dictates the convective available potential energy (CAPE) [1]. Across the narrow North Pacific ITCZ between 5–15°N, for a given rain rate, there are approximately twice as many thin cold clouds where the underlying large-scale SST is about 1.0–1.5 °C higher, with a greater fraction of low- and mid-level clouds where mean SST gradients are larger [2,3]. SST gradients were posited by Lindzen and Nigam [4] to determine airflow and hence low-level convergence and convection; moisture boundary layer convergence was later confirmed to be a cause of convection by Back and Bretherton [5].

Closely connected to the vertical distribution of tropical clouds is the vertical structure of pressure vertical velocity ($\omega$). A top-heavy profile, such as over the western tropical Pacific, in which ascent peaks near 400 hPa, in conjunction with higher upper-tropospheric humidity, may help sustain more cirrus clouds than in regions with bottom-heavy $\omega$-profiles [6], e.g., the eastern Pacific. The vertical and horizontal cloud structure differences over the western tropical Pacific contribute to clouds having a significantly smaller net TOA cooling effect versus cloud profiles in the eastern Pacific [2].

Over the broader western tropical Pacific, the local Hadley circulation also migrates seasonally, as the maximum solar insolation changes over the annual cycle. During a year, SST maximizes on the poleward side of the equator in the summer hemisphere, shifting the SST distribution, SST gradients, and the location and strength of the low-level convergence. The corresponding location of maximum ascent at 500 hPa is just north of the equator during July, and just south of the equator in January [7]. SST hot spots, which are defined as 15-day (or longer) SSTs that exceed 30 °C over a large region (e.g., 20° longitude x 10° latitude), have recently been studied by Kubar and Behrangi [8,9], and have a propensity to form in the near-equatorial SPCZ region in December (and in some years a secondary maximum in March or April). Hot spots are followed by maximum low-level convergence, upward motion, and thick high and anvil cloud cover one to two months later in January and February. Shallow upward motion in the SPCZ region, in contrast, peaks in July. Thus, the migration of the western Pacific Hadley circulation is associated with changes through the year of the location of maximum SST and meridional SST gradients. The tropical west Pacific, which we refer to henceforth as the TWP, is symmetric about the equator (30°S–30°N; 120°E–155°W), is fortuitous for studying tropical cloud feedbacks as the SST mode does not change with mean TWP SST, but the shape of the SST distribution becomes more or less peaked, implying less or more SST variance, respectively. This allows study of SST-cloud–precipitation relationships as a function of different domain SST gradients, which drive changes in local convergence and upward motion patterns and may drive local cloud patterns but not total cloud cover. The TWP SST will be shown to be particularly useful for examining a range of convective-onset SSTs; Johnson and Xie [10] have demonstrated how onset SST varies with the mean climate state.

High cloud feedbacks against changes in SST have long been a topic of research, but the question of whether or not higher SST solely drives stronger convection and generates more upper-level cloud locally, or also in a largely adjoining region beyond the strongest updrafts, has been an area of active study for decades. Many studies have documented the increase of total high cloud amount with local SST [8,9,11–16], with an explicit decrease of deep clouds over the very highest SSTs noted in particular by [8,9,17]. That tropical precipitation is heaviest in regions of high SST has a particularly long history in the literature, including from early studies of [12,18–21] and others within Neelin and Held [12]. That rain rates are particularly high in the Central and East Pacific ITCZ, however, suggests the importance of SST gradients, low-level convergence, and the contribution to total rain from shallower convection, where underlying SSTs are lower compared to the TWP. Thus, SST by itself may be insufficient in areas or during times of the year in predicting precipitation maxima when low-level convergence is strong [22].

Whether the high cloud-SST relationships hold on a larger scale such as the TWP has been an area of active research in terms of cloud feedbacks. A study by Lindzen et al. [23] identified possible reduction of upper-level cirrus cloud fraction normalized by cumulus coverage with increasing SST over the broad western Pacific, and called the apparent mechanism the Iris Effect, with an analog to the iris of an eye which opens more, in conjunction with a contracting pupil, with increasing light [23]. The decrease of semi-transparent cirrus clouds allows additional outgoing longwave radiation, a negative cloud feedback with SST. While questioned by Hartmann and Michelsen [24] by demonstrating the original study's inclusion of subtropical clouds disconnected with deep convective cloud areas, Fu et al. [25] also pointed out an assumption of the original Iris Effect of constant low cloud cover with TWP SST. This may seem dubious given the strong negative feedback locally of SST and low cloud cover over much of the low latitudes [26]. Another assumption of the original Iris Effect study was of a constant low cloud albedo of 0.4 and a high cloud albedo of 0.2 to match the mean tropical albedo observations from Earth Radiation Budget Experiment (ERBE) of 0.3. Thus, the Iris Effect suggests that a decrease of high clouds would expose a greater area of the tropics to the strong cooling effect of low clouds. With Multi-Angle Imaging Spectroradiometer (MISR) data and collocated CERES TOA radiative flux data, we have the opportunity to investigate and quantify the relationship and sensitivity of cloud albedo both as a function of convective strength, and the entire TWP SST. Our construction of cloud height-cloud albedo ($Z$-$\alpha$) histograms for TWP SST quintiles and local rain rates will shed insights about each of the assumptions made in [23]. Indeed, Lindzen et al. [23] calculated a strong negative feedback factor associated with the Iris Effect.

The Thermostat Hypothesis, first investigated by Ramanathan and Collins [27] over 25 years ago using data from the 1987 El Niño, postulates that high anvil clouds thicken as SSTs warm. The clouds do so, as the authors explain, in response to a "super greenhouse effect" at very high SSTs, in which the atmospheric greenhouse effect of trapping outgoing longwave radiation exceeds the rate at which radiation can be emitted at the surface. At high temperatures, the authors describe the strong greenhouse effect as due to high total-column water vapor, high middle- and upper-troposphere $H_2O$ concentration, and changes in the lapse rate. The formation of thick high clouds act as a negative feedback, a regulator of tropical SSTs. However, the greater abundance of thin, cold cloud where SSTs are warmer [2], calls into question whether a Thermostat Effect exists if precipitation is normalized, or whether the correlation of more thick clouds with SST is primarily due to cloud optical depth scaling with precipitation rate.

While a number of studies have found either countering or weak evidence of an Iris Effect [28–30], other studies in recent years have demonstrated support for the effect. Igel et al. [31] used a cloud-object approach with CloudSat data to show both a shrinkage of anvil width and an increase in physical anvil depth with SST per cloud object, the latter finding a nod to the Thermostat Hypothesis. While the broader implications about high cloud behavior in a large basin is not explicitly investigated in Igel et al. [31], recently Choi et al. [32] have studied the Iris Effect using updated TRMM and A-Train satellite data, and find a reduction in cirrus fraction with mean TWP SST, with a CF slope of −0.0745/(°C SST). However, their use of the MODIS cirrus fraction product may provide a mixed signal of middle-level convection and truly high clouds, since the pressure threshold for that variable is 500 hPa, which may correspond to the near-mean detrainment level of congestus clouds. A decrease in mid-level congestus convection was proposed as a possibility in Rapp et al. [29] when using warmer brightness temperature thresholds to quantify local cloud relationships with SST. In our study, we demonstrate that low and middle clouds in the TWP behave very differently with mean TWP SST from clouds associated in the upper troposphere. A decrease in mid-level clouds would also have a similar TOA radiative effect as a loss of low clouds; the total TOA cloud albedo would decrease, and the likely effect would be net TOA warming since the LW effect of mid-topped clouds is far weaker than for high-topped clouds.

Given the ongoing concerns and uncertainties regarding local and total cloud behavior as a function of TWP SST, and the opportunity to use MISR with CERES data to discriminate different high-topped and other cloud types with rain rates from TRMM, our objectives are thus as follows:

(1)　Use the TWP domain to test how SST distribution properties such as kurtosis change with domain-mean SST, and how these changes influence local cloud-precipitation-SST relationships and the local Hadley Circulation. Key TOA cloud effects examined are cloud albedo and net cloud forcing.

(2)　Quantify the importance of local SST gradients versus SST in determining total high cloud amount locally and away from the highest SSTs, as well as the convective SST onset as a function of large-scale mean SST.

(3)　As in Kubar et al. [2], investigate the cloud structure of precipitating systems by constructing Z-$\alpha$ histograms; how do these change with convective strength and domain-mean SST?

(4)　Investigate the extent to which local and domain-wide TWP high cloud changes can be attributed to associated $\omega_{500}$/large-scale dynamics changes, versus those that may be related to either an Iris Effect or thermodynamic effects associated with horizontal SST structure changes.

## 2. Observational and Reanalysis Datasets

In this study, we take a multi-sensor and multi-dataset approach to quantify how both local and domain-wide cloud properties, including their fractional coverage, vertical distribution, and TOA cloud albedo, vary with local SST and rain rate, as well as how these local relationships are modified by domain-wide mean SST and horizontal SST structure. Most of this study focuses on precipitating-only clouds. We furthermore assess how low-level divergence and mid-tropospheric vertical velocity vary with local SST against different domain-mean SSTs, to construct a comprehensive picture about the interactions between tropical clouds, local and large-scale SST, and circulation properties in one of the warmest large ocean regions of the globe. While the primary instrument used is MISR vertical profiles of cloud fraction and grid-scale TOA cloud albedo, for which we could draw many of the conclusions found in this study, we also use complementary CERES data for cloud optical depth and TOA net cloud forcing, TRMM for high-resolution precipitation, and ERA-Interim reanalysis data for key dynamics variables and SST.

### 2.1. MISR

The core observational dataset is the Level-3 (L3) Multi-Angle Imaging Spectroradiometer (MISR) daily products, both the 0.5° × 0.5° cloud fraction by altitude (CFbA) product, as well as the 1° × 1° TOA albedo product. Unlike other passive satellites, MISR directly measures geometric height from multiple directions and needs no additional temperature profile information [33,34]. Recent improvements in the CFbA product include far fewer retrievals of "NoRetrieval" heights of clouds, due to improved height coverage (Moroney et al. [35]). The MISR instrument flies at an altitude of 705 km on a descending orbit onboard the EOS-Terra satellite, with global coverage every two to nine days, depending somewhat on latitude. We use two years of data from October 2002 through September 2004 for an intensive analysis of cloud vertical, horizontal, and reflectivity properties as a function of local and domain-scale states of the TWP, and how these are related to the horizontal distribution of SST, and tropical cloud properties and feedbacks. We construct domain-averages of the entire TWP for each day of the two years studied (October 2002 through September 2004), as well as the northern (0°–30°N) and southern (0°–30°S) portions of the TWP, and save all pertinent grid-scale variables to examine relationships as a function of TWP SST quintiles.

We note that Davies et al. [36] discovered artefacts associated with the MISR time series of global effective cloud top height during the early part of the record, associated with a different equatorial crossing time of the Terra satellite. Specifically, the equatorial crossing time of Terra was about ~10:45 a.m. local time from March 2000 through March 2002, and ~10:30 a.m. local time thereafter. This later time during the first two years results in more sun glint at the eastern edge of retrievals during that period, and more sun glint has the impact of increased detection of high-level clouds. After March 2002, Terra's crossing time stabilized, ruling out subsequent instrument drift/error as a cause of trends thereafter. Fortuitously, our analysis period from October 2002–September 2004, chosen to coincide

with the availability of CERES joint Aqua/Terra data, skirts the later Terra satellite equator-crossing time issue. Towards the end of this study, we do briefly quantify the possible effect of a moderate warm ENSO event (2002–2003; McPhaden [37]) compared to a neutral year (2003–2004) by splitting the data into two 12-month chunks.

To estimate top-of-atmosphere albedo, MISR uses a "multi-angle observing strategy" with its nine pushbroom cameras by measuring the intensity of reflected sunlight, one at nadir and four in the forward along-track and aftward directions of nadir at 26.1°, 45.6°, 60.0°, and 70.5° [34]. The track length of 2800 km has a swath width of ~400 km. The retrieval of albedos from all nine angles rather than only one is a unique and important feature of MISR [35], as is its high spatial resolution. Previously, a near-global (75°S–75°N) comparison was performed between MISR and CERES albedos for overcast ocean scenes, with very small differences between the two instruments of approximately 3.8% [34]. In our study, we screen out land grids to study oceanic convection over the warm western tropical Pacific.

We use the expansive albedo average, for which the L3 grids are averaged at $1° × 1°$ from the native L2 measurements at 35.2 km [38]. The average as defined here for the $1° × 1°$ grid is the reflected flux divided by the average solar insolation, but effectively this is very similar to the direct average of L2 albedos. We then calculate the cloud albedo, $\alpha_{cloud}$, as

$$\alpha_{cloud} = [\alpha_{total} - (1 - CF)\alpha_{clr}]/CF \tag{1}$$

In (1), $\alpha_{total}$ is the measured all-sky albedo from MISR, and CF is the total cloud fraction for each $1° × 1°$ grid from MISR. We briefly examine $\alpha_{cloud}$ from CERES as well towards the end of this study; at that stage we also use the corresponding total CF from CERES. The clear-sky albedo, $\alpha_{clr}$, is only available from CERES; $\alpha_{clr}$ varies modestly over the ocean compared to all-sky albedo. In the text henceforth, the definition and use of cloud albedo comes from (1). The cloud albedo, or reflectivity, represents the shortwave cooling effect of clouds, and both cloud cover and the albedo contribute to the strength of this effect. Using cloud albedo, rather than all-sky albedo, allows us to compare against other studies such as Chae and Sherwood [33]. While cloud optical thickness is fundamentally related to cloud albedo, which we explore as well in this study against CERES cloud/radiative flux data, cloud albedo is also related to the microphysical properties of clouds. A cloud with the same optical thickness has a higher (lower) albedo when the mean effective radius is smaller (larger), though we save explorations of microphysical relationships and convection strength and structure for future work.

MISR contains three albedo variables—expansive, restrictive, and local—each of which are available at four spectral bands, centered at 446 nm ± 21 nm, 558 nm ± 15 nm, 672 nm ± 11 nm, and 866 nm ± 20 nm, as well as a broadband approximation. For this study, we use the average $1° × 1°$ expansive broadband albedo, which is the albedo that would be measured at an altitude of ~30 km, and incorporates reflection from the entire viewable scene; this albedo product is described as providing the best estimate for TOA energetics in the region. Loeb et al. [39] computed an approximate 4% error in SW radiances in the narrow-to-broadband coefficient conversion. Offline, we make comparisons of the three TOA albedo products by making annual maps, which underscores good qualitative similarity between the three products. As explained by Sun et al. [34], while the restrictive albedo is most similar to the ERBE or CERES albedo definition, both the expansive and restrictive albedos use angular distribution models (ADMs).

Two of the most critical components of climatological variables examining the climate effect of clouds are both cloud fraction and cloud top height, which are contained in the MISR Cloud Fraction by Altitude (CFbA) $0.5° × 0.5°$ product, with 41 vertical levels and 500 m vertical sampling. The L3 cloud fraction is produced from the L2 1.1 km nadir cloud fraction, with a pixel considered cloudy if it is high-confidence or low-confidence cloudy. If there is sun glint contamination, the first forward camera cloud field is viewed, followed by the first aftward camera if the forward camera is contaminated. Each of the subsequent angular cameras are checked until a glint-free pixel is found.

The L3 cloud top height is derived from the median L2 17.6 × 17.6 km regions, which themselves are derived from the native 1.1 km spatial resolution cloud top height, using MISR's stereoscopic technique and additionally corrected using wind retrievals from the "best quality category" [40]. When no height is known but a cloud fraction is still available, we include the missing portion and scale the known CF(z) in a grid by the missing portion. The corresponding cloud fractions and heights are then projected on 0.5° × 0.5° grids, which we aggregate into 1° × 1° grids in order to match expansive albedo grids. The joint distribution of cloud fraction and height from MISR represents the current highest-resolution CFbA product from any passive instrument [40]. We assign each of the possible 41 cloud height bins in each 1° × 1° grid the same retrieved $\alpha_{cloud}$, since individual unique albedos are not available for MISR L3 at the subgrid scale. These are only available for L2 data, which we do not use for this study. This method is somewhat analogous to Kubar and Hartmann [3]; in that study the coarser 25 km footprint of AMSR-E rain rates was matched with the higher-resolution retrievals of CloudSat cloud profiles, so that each of the clouds were assigned the same rain rate in the coarser AMSR-E box. Similarly, all clouds within each MISR grid are assigned the same $\alpha_{cloud}$. In future work, we plan to quantify the sensitivity of this assumption at the 1° × 1° scale versus using albedos at each native pixel.

We conclude here with a few other aspects of the MISR configuration; despite it being a passive sensor, MISR much more effectively senses low-topped clouds than other passive sensors. Marchand et al. [41] examined histograms of cloud top height from MISR, ISCCP, and MODIS, and demonstrated that MISR data "provide more accurate retrievals of cloud top height for low-level and midlevel clouds, more reliable determination of midlevel clouds from other clouds, and better detection of trade cumulus". We do not infer that MISR retrievals are perfect; indeed the authors note the slightly better sensitivity of ISCCP versus MISR for very optically thin clouds, and the utility of MODIS for optically thick clouds. However, we will be able to test the assumption made by Lindzen et at. [23] of constant tropical low cloud cover and albedo [25].

A related general challenge of passive sensors is multilayered clouds; global cloud properties, biases, and differences among sensors were assessed by Stubenrauch et al. [42] in conjunction with the Global Energy and Water Cycle Experiment (GEWEX) Radiation Panel. Depending on the spectral channels, geometry, angles, and resolution, active and IR methods emphasize cloud tops of transparent cirrus clouds when overlying low-level clouds, IR-VIS methods reflect the mean radiative height of the two clouds, and visible-only methods, e.g., MISR, preferentially display the low-level cloud properties. The authors noted that unlike other passive satellites, MISR accomplishes this through the judicious use of multiple view angles of solar reflectances; the authors note a global low-level cloud amount of 60% as seen by MISR, slightly higher than CALIPSO-GOCCP (57%). Fortuitously, for our study, this enables variations of low-level clouds to be quantified as a function of large-scale SST and local SST and rain rate.

## 2.2. CERES (Clouds and the Earth's Radiant Energy System)

Top-of-atmosphere radiative flux data are essential to quantifying the net radiative effect of clouds, and the CERES dataset greatly enhances the MISR data central to this effort. As in Kubar and Behrangi [8,9], we use the CERES 1° × 1° SYN1DEG product, which contains TOA and profile fluxes [43] from the native 20-km spatial resolution at nadir. The CERES dataset used here is from the combination of data from the Terra and Aqua platforms, thus representing the daytime average (local equatorial noon), slightly later from the MISR equatorial morning-only overpass (10:30 a.m.). We briefly summarize the implications of the differences in temporal characteristics of MISR, CERES, TRMM, and ERA-Interim in Section 2.3. In addition to computing the $\alpha_{cloud}$ as for MISR above, we also use the shortwave (SW) and longwave (LW) TOA fluxes to compute the net radiative effect, or forcing, of clouds. We thus need CERES data since TOA radiative fluxes are not retrieved by MISR. The SYN1DEG product also has MODIS derived and geostationary cloud products, including cloud area fraction, cloud optical depth (not available from MISR), cloud effective height, and cloud effective temperature.

We use these in construction of the net cloud forcing binned by Z-$\alpha_{cloud}$, to complement the MISR Z-$\alpha_{cloud}$ histograms of cloud fraction. Visible optical depth is only available from MODIS/CERES.

As discussed earlier, MISR uses ADMs at the local albedo level for the expansive albedo retrievals; uncertainties of albedos from both CERES and MISR stem from these ADMs, though validation studies have confirmed radiative flux accuracy improvements of CERES ADMs [44,45]. From Smith et al. [46], CERES errors of TOA absorbed solar radiation (ASR) and outgoing LW radiation are 1 W m$^{-2}$ and 2.4 W m$^{-2}$, respectively, which correspond to 0.5% and 1%. Both Loeb et al. [39] and Sun et al. [34] quantify the error of converting narrow-to-broadband MISR albedos at ~2%; the RMS difference between MISR and CERES albedos due to ADMs is only 3.8%, these conversion regressions are much smaller than earlier studies and indicate excellent consistency between CERES and MISR.

### 2.3. TRMM 3b42

Following Kubar and Behrangi [8,9], precipitation observations are critical to this study, and come from The Tropical Rainfall Measurement Mission (TRMM) Multisatellite Precipitation Analysis (TMPA V7; Huffman et al. [47]), which provided precipitation estimates for more than 17 years, starting in 1997, from multiple satellite platforms as well as gauge data at a high horizontal resolution of 0.25° × 0.25°. Daily-averaged data from original three-hour estimates are used in conjunction with cloud property data to assess how the horizontal and vertical cloud structure and cloud radiative effects depend on rain rates, a proxy for convective intensity [2]. Temporally-averaged daily data from CERES, TRMM, and ERA-Interim are effectively later than the ~10:30 a.m. LT equatorial crossing time of the Terra satellite (MISR), but this may be a fairly minor concern given the small diurnal cycle of tropical convection during the day over the ocean [33,48].

Additionally, we test how different cloud types vary with local rain rate, and how these relationships change with domain-mean SST. Though there may be underestimation of light precipitation in subsidence regions from microwave (MW) products of 10% or more [48], rain rate estimates from TRMM are considered reliable over tropical oceanic regions, as shown in Behrangi et al. [49,50].

### 2.4. ERA-Interim

Sea-surface temperature (SST) data at 1° × 1° come from ECMWF Reanalysis (ERA-Interim; Simmons et al. [51]), for which we average for daily means at the grid- and domain-scales. Additional information about the circulation comes from pressure vertical velocity (ω) at 850 hPa, 700 hPa, and 500 hPa, as well as low-level divergence (DIV) at 1000 hPa, 925 hPa, and 850 hPa. Profile reanalysis data of large-scale dynamics from ERA-Interim have been used in Kubar et al. [26] and Kubar and Behrangi [8,9]. Relatively small differences exist between the different ERA-Interim horizontal resolution products.

## 3. Cloud Definitions and Domain-Choice

The Tropical West Pacific (TWP) is a large warm ocean region between 120°E–155°W and 30°S–30°N, where middle and deep convective, anvil, and cirrus clouds are prolific. Near-equatorial SSTs tend to be the warmest on the globe in a large region near and just south of the equator within this domain, and Kubar and Behrangi [8,9] described how SST hot spots, in which a 20° longitude by 10° latitude mean ocean region exceeds 30 °C, are most favorable just south of the equator and just west of the International Dateline. This subdomain and identification of hot spots was studied earlier by Waliser [52]. The focus of the analysis in Kubar and Behrangi [8] was the time series of hot spots and the subsequent lag of deep convection, best described using a simple predator-prey model, with convection the predator and SST the prey, considered the source of convection. In Kubar and Behrangi [9], the focus is the spatial structure, large-scale circulation, and evolution between high SST and deep convection as a function of hot spot phase, with an important finding of cirrus clouds being collocated with the highest SSTs and hot spot features, both temporally and spatially. The focus of Kubar and Behrangi [9] is also of teleconnections associated with SPCZ hot spots, and how changes

associated with the Walker Circulation might be associated with hot spot formation. Here, we focus on the large TWP domain to examine how the clouds both locally and in the aggregate respond to changes in the large-domain SST, which also modulates the north–south SST gradients.

We use a classification of high-topped clouds in this study similar to Kubar et al. [2] and Kubar and Behrangi [8,9], except here cirrus, anvil, and thick high clouds are determined from TOA cloud albedo rather than visible optical depth ($\tau$). TOA cloud albedo is fundamentally related to $\tau$, and we use collocated MISR cloud albedo and CERES high cloud $\tau$ to determine the corresponding albedos to cirrus ($0 < \tau < 5$), anvil ($5 < \tau < 30$), and thick ($\tau > 30$) clouds. For the purposes only of Figure 1, we consider any cloud with a top > 7 km a higher cloud, since effective cloud heights from CERES are lower than MISR cloud top heights. This is the only true joint distribution in this study in which we use time-collocated MISR cloud top heights and CERES effective heights; most of the rest of the text is MISR cloud top height-centric. In the rest of the text, low clouds have tops between 0–4 km, middle clouds 4–9 km, and high-topped clouds tops > 9 km, all of which are based on MISR cloud top height data. This categorization of high clouds is similar to Kubar and Hartmann [3], with such height cutoffs defining the natural modes of cloud tops. Other studies have also characterized a trimodal distribution of tropical cloud heights, for example in Johnson et al. [53] and Chae and Sherwood [33].

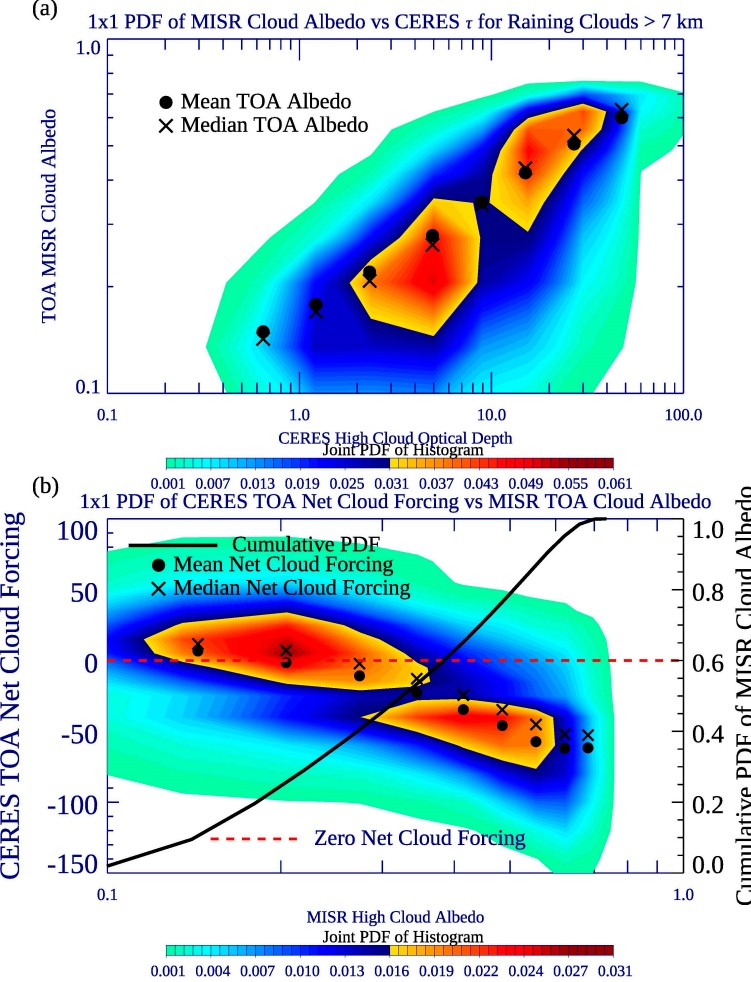

**Figure 1.** (**a**) Joint PDF of $1° \times 1°$ TOA cloud albedo versus CERES high cloud optical depth $\tau$ for raining clouds (as defined by collocated TRMM grids) for grids in which the MISR cloud fraction > 0.2 above 7 km, and the high-topped cloud fraction>low-topped cloud fraction. Mean and median values are shown for each bin. (**b**) Same as (**a**), except joint PDF of CERES net cloud forcing versus MISR high cloud albedo. The red dashed line denotes zero net cloud forcing, and the black line depicts the cumulative PDF.

To determine appropriate albedo thresholds for cirrus, anvil, and thick high clouds, we construct a joint PDF of collocated TOA cloud albedo from MISR and high cloud optical depth from CERES for raining grids, the latter determined by TRMM. The high cloud optical depth is a product of the CERES SYN1DEG dataset, and we require that the cumulative MISR CF at or above 7 km for each aggregate $1° \times 1°$ composite profile be greater than 0.2, and also that the total CF > 7 km is greater than the total CF below 7 km. This height inclusion is a more lenient definition that includes some upper-middle clouds in the rest of this study, but is typically above the freezing level.

The joint histogram of TOA MISR $\alpha_{cloud}$ and CERES high cloud $\tau$ is presented in Figure 1a, with both the *x*-axis and *y*-axis shown on logarithmic scales. TOA cloud albedo is strongly correlated with CERES High Cloud $\tau$, with the mean and median albedo shown as well for each $\tau$ category. For a visible $\tau$ of ~ 5, the mean TOA albedo is slightly under 0.3, and for an optical depth of 30, while the mean or median albedo is just above 0.5, the mode (bright orange contours) is closer to 0.6. We use 0.3 and 0.6 for our two cutoff albedos, with cirrus clouds henceforth defined as profiles in which grid-scale $\alpha_{cloud} < 0.3$, $0.3 < \alpha_{cloud} < 0.6$ for anvil clouds, and $\alpha_{cloud} > 0.6$ for thick clouds. As a note, the 0.3 cloud albedo threshold is the same as used in Chae and Sherwood [33] to distinguish thin from thicker clouds, with the thicker cloud category described by the authors as either convective clouds or thicker anvil clouds.

In terms of the lack of very optically thin clouds, there are several reasons why we might anticipate their absence. The grids selected here (and for much of this study) are for raining grids only, perhaps reducing the expectation of few optically thin clouds. In Figure 1a, 0.2% of the entire joint histogram encompasses clouds with visible optical depths in the 0.2 to 0.4 category. Moreover, however, the general minimum $\tau$ detection limit of passive imagers, depending on algorithm, tends to be between 0.1 and 0.3 [54]. An active instrument, such as the CALIPSO lidar, can see clouds with visible optical depths as small as 0.01, and could be a beneficial complement, but given its sparsity relative to MISR (which itself is less voluminous in terms of swath size compared to CERES or TRMM), is not considered in this intensive two-year study.

One distinguishing factor of high cirrus clouds in Kubar et al. [2] is that they have a net TOA warming effect on the climate; indeed, these are the only high clouds that tend to, as the longwave warming effect dominates the shortwave cooling effect more as cirrus clouds become colder. Figure 1b demonstrates the TOA net cloud forcing versus MISR high cloud albedo for screened, raining grids as for Figure 1a. The TOA cloud forcing mode is slightly to moderately positive when MISR albedos < 0.3, consistent with Kubar et al. [2] and Fu et al. [25]. More recently, using CALIPSO and CloudSat measurements, Hong et al. [55] assessed the radiative effects globally of ice clouds, and show that ice clouds with $\tau < 4.6$ have a net TOA warming effect, with the largest contributors from clouds with $\tau$ ~ 1.0. Clouds thinner than this thus have a smaller contribution to net TOA cloud forcing. That assessment matches the net cloud radiative forcing calculations made in Kubar et al. [2] using the Fu and Liou [56] delta-four-stream, k-distribution scheme radiation model. In [2], a fully overcast thin cold cloud with $\tau = 1$ has a peak net cloud effect of +94.3 W m$^{-2}$. Our results here are broadly consistent; the mean TOA cloud forcing maximizes when TOA albedo ~0.15, which from Figure 1a corresponds to slightly less than $\tau$ ~ 1. As we have no requirement of overcast grids in Figure 1, a smaller positive net cloud forcing would be expected.

## 4. Results and Discussion

### 4.1. Distribution of SST, Clouds, Rain Fraction, Albedo, and Circulation vs. Large-Scale SST

In studies using the TWP to assess a possible Iris Effect of high-topped clouds [32], the horizontal distribution of SST and corresponding large-scale circulation has largely either been overlooked, or else neglected, with the main driver considered the domain-mean TWP SST. Increases in TWP SST are postulated to concentrate outflow clouds and reduce stratiform rain rates due to higher precipitation efficiency. In this study, we wish to understand how the horizontal SST structure changes as a function

of TWP SST, and how such changes drive the regional circulation, which are directly linked to observed cloud changes as the domain-mean SST changes. Thus, we investigate to what extent dynamics can explain Iris-like cloud changes or non-Iris changes.

In Figure 2 we present maps of SST (left column), high cloud fraction (all clouds with tops > 9 km, middle column), and low+middle cloud fraction (all clouds with tops below 9 km, right column). Each row represents either the SST or cloudiness when domain-mean TWP SST ($SST_{TWP}$) falls within one of five SST quintiles, from coldest (top) and warmest (bottom). Thus, the first row is for times in which 0th < $SST_{TWP}$ < 20th percentiles, the second row 20th < $SST_{TWP}$ < 40th percentiles, the third row 40th < $SST_{TWP}$ < 60th percentiles, the fourth row 60th < $SST_{TWP}$ <80th percentiles, and the bottom row 80th < $SST_{TWP}$ < 100th percentiles. We employ this partitioning through much of this study, which serves as a useful test for which to examine how local properties and relationships vary under the background of different large-scale SSTs. Later, we incorporate local precipitation and composite clouds and dynamics as a function of local rain rates and SST for TWP SST quintiles.

As $SST_{TWP}$ increases, the total north-south extent of the warm pool, e.g., the SST contour > ~27 °C, increases, as does the size of maximum SST area, but the maximum SST value near the equator does not increase. Instead, for any of the five SST states, the axis of highest SSTs tends to be oriented from northwest to southeast. This is similar to the contours of high cloud fraction (middle column). For the lowest $SST_{TWP}$, very low values of high CF, e.g., CF < 0.04, are in the northernmost part of the domain—e.g., the white area (Figure 2b)—but the white area diminishes to nearly zero in the second and third rows. Then, high clouds become scarce in the southwestern part of the domain, just north of Australia, for the second warmest and warmest TWP regimes, as the warm pool expands further north rather than south. There is no a clear expansion or contraction of high cloud area with $SST_{TWP}$, but there is a much more clear and robust decrease of low+mid CF with increasing $SST_{TWP}$ (right column), as more clouds transition from low or middle modes to deep modes with a broader area of higher SSTs. Furthermore, the size of the coolest SST area, where low clouds are most abundant, decreases with increasing TWP SST quintile.

Figure 3 presents an analogous display, but for rain fraction (left column), the cirrus portion of raining high clouds (middle column), and TOA albedo (right column) of raining grids. As the warm pool shifts from being more southern-hemisphere centric to northern hemisphere-centric (top to bottom), the maximum rain fraction area also transitions from more southern to northern hemisphere area, though in general the rain fraction is always high locally along the SPCZ and northern ITCZ. The more obvious change is in the composition of high clouds; as $SST_{TWP}$ warms, the portion of high clouds that are cirrus clouds increases. The cirrus portion of high clouds is largest in general in the NH, but then a second maximum develops along the central equatorial region for particularly the bottom three rows. Finally, total cloud scenes tend to become less reflective (right column) with increasing $SST_{TWP}$, and the conditional cirrus cloud fraction tends to be anticorrelated geographically with TOA albedo.

In Figure 4, we present $\omega_{500}$ (left column) and $DIV_{850}$ (right column) for scenes screened for rain, and generally ascending motion is stronger in the mid-troposphere near the highest SSTs when the domain is coolest. For the lowest $SST_{TWP}$ (top), there is modest subsidence between nearly the entire zonal region of ~15°–20°N, with the subsidence area generally smaller as $SST_{TWP}$ warms. This pattern suggests a stronger regional Hadley Circulation when $SST_{TWP}$ is coolest, owing to the larger north–south SST gradients. This is even more clear for $DIV_{850}$; there is moderate low-level divergence to the north of 10°N, but the strength of this divergence decreases with $SST_{TWP}$. The local near-equatorial maximum of convergence is also somewhat stronger when $SST_{TWP}$ is cooler.

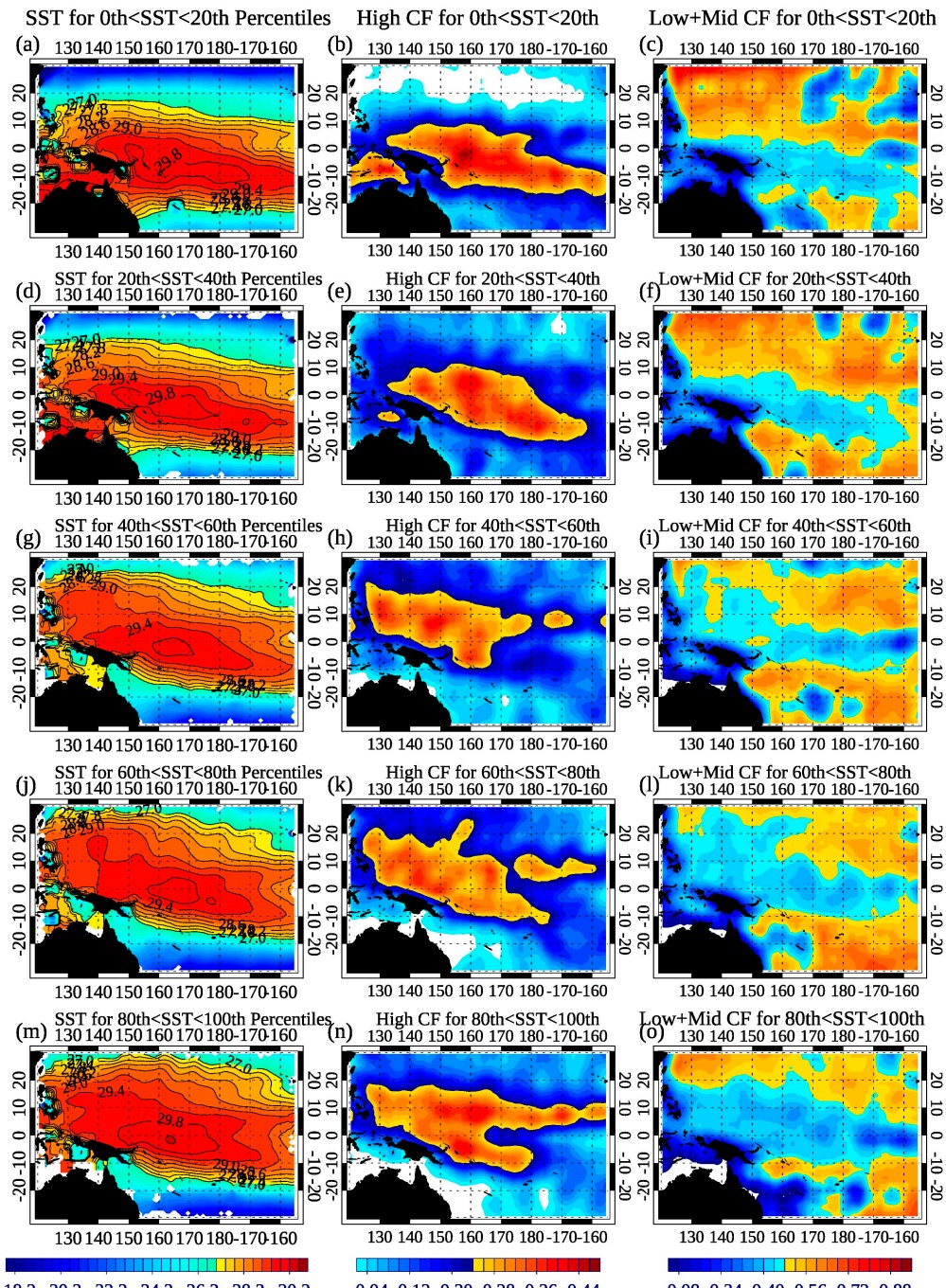

**Figure 2.** Map of mean (**a**) SST, (**b**) MISR high cloud fraction (tops > 9 km), and (**c**) MISR low + middle cloud fraction (tops < 9 km) during times when the mean TWP SST is in the first quintile (26.70 °C < $SST_{TWP}$ < 27.09 °C). (**d**–**f**): Same as (**a**–**c**), except during times when the mean TWP SST is in the second quintile (27.09 °C < $SST_{TWP}$ < 27.29 °C). (**g**–**i**): Same as (**d**–**f**) except for times when the mean TWP SST is in the third SST quintile (27.29 °C < $SST_{TWP}$ < 27.41 °C). (**j**–**l**): Same as (**g**–**i**), except for times when mean TWP SST is in the fourth quintile (27.41 °C < $SST_{TWP}$ < 27.51 °C). (**m**–**o**): Same as (**j**–**l**), except during times when mean TWP SST is in thefifth SST quintile (27.51 °C < $SST_{TWP}$ < 27.88 °C). The MISR CF in the middle and right columns is conditional for daily raining grids only.

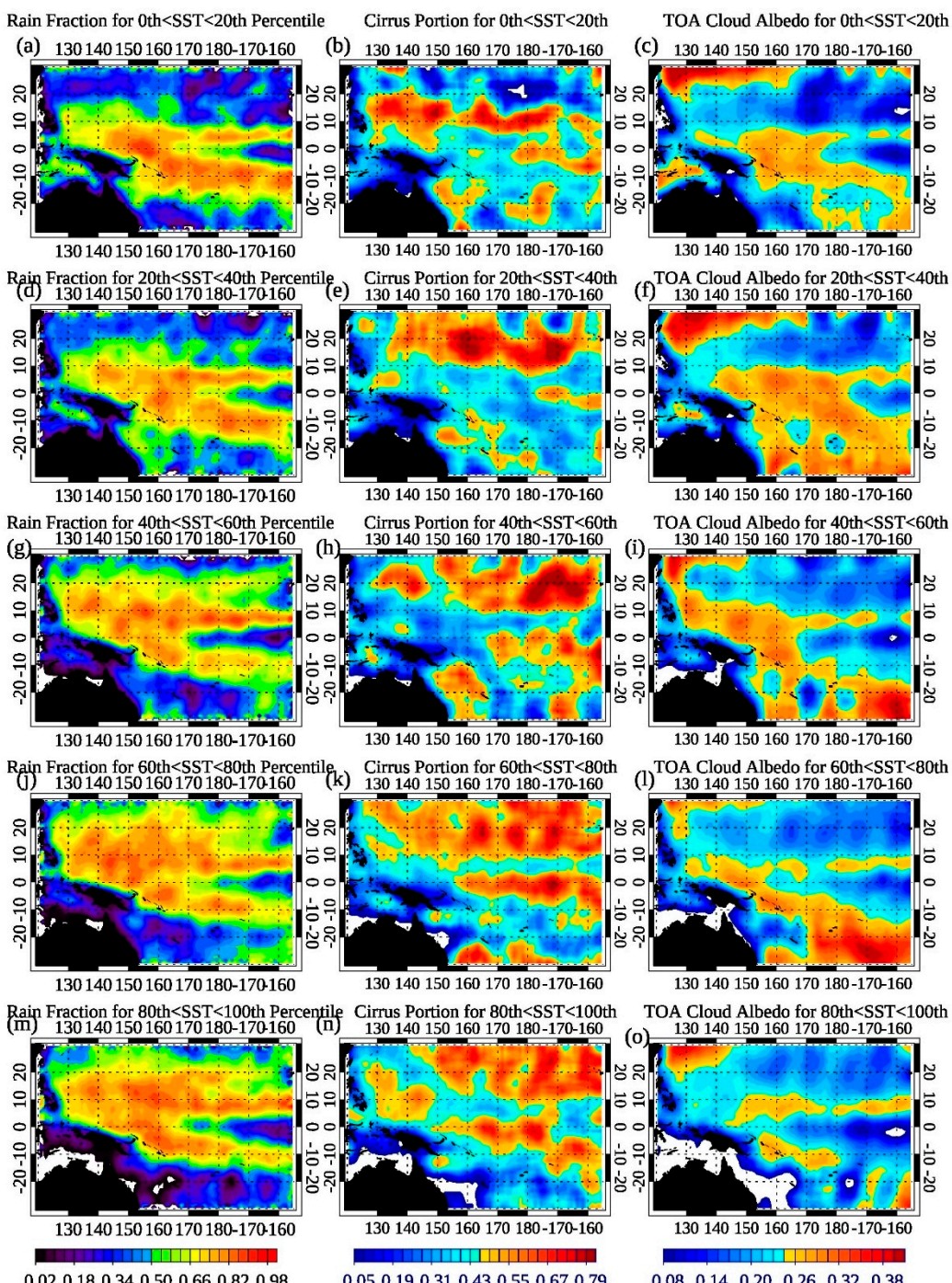

**Figure 3.** (**a**) Fraction of raining grids, (**b**) cirrus fraction of total high cloud fraction, and (**c**) TOA cloud albedo during times when the mean TWP SST is in the first SST quintile (26.70 °C < $SST_{TWP}$ < 27.09 °C). (**d–f**): Same as (**a–c**), except during times when the mean TWP SST is in the second SST quintile (27.09 °C < $SST_{TWP}$ < 27.29 °C). (**g–i**): Same as (**d–f**), except during times when the mean TWP SST is in the third quintile (27.29 °C < $SST_{TWP}$ < 27.41 °C). (**j–l**): Same as (**g–i**), except during times when the mean TWP SST is in the fourth quintile (27.41 °C < $SST_{TWP}$ < 27.51 °C). (**m–o**): Same as (**j–l**), except during times when the mean TWP is in the fifth quintile (27.51 °C < $SST_{TWP}$ < 27.88 °C). The MISR cloud properties in the middle and right columns are conditional for daily raining grids only.

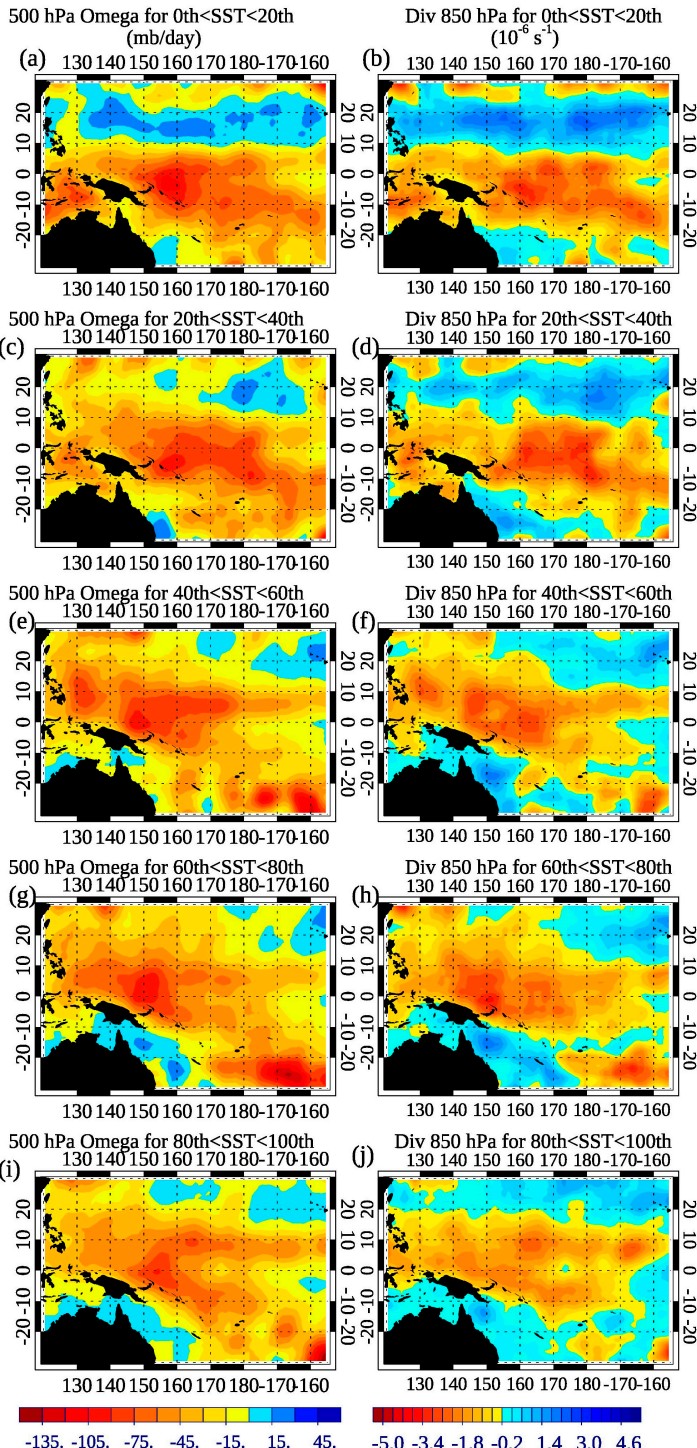

**Figure 4.** (**a**) $\omega_{500}$ (mb/day) and (**b**) divergence at 850 hPa ($Div_{850}$) ($1 \times 10^6 s^{-1}$) during times when the mean TWP SST is in the first quintile (26.70 °C < $SST_{TWP}$ < 27.09 °C). (**c,d**): Same as (**a,b**), except during times when the mean TWP SST is in the second quintile (27.09 °C < $SST_{TWP}$ < 27.29 °C). (**e,f**): Same as (**c,d**), except during times when the mean TWP SST is in the third quintile (27.29 °C < $SST_{TWP}$ < 27.41 °C). (**g,h**): Same as (**e,f**), except during times when the mean TWP SST is in the fourth quintile (27.41 °C < $SST_{TWP}$ < 27.51 °C). (**i,j**): Same as (**g,h**), except during times when the mean TWP SST is in the fifth quintile (27.51 °C < $SST_{TWP}$ < 27.88 °C). All panels are conditional for daily raining grids only.

## 4.2. Relationships vs. SST$_{local}$ as a Function of Mean SST$_{TWP}$

While the aforementioned maps highlight important aspects of how SST, clouds, and large-scale dynamics vary across the TWP with SST$_{TWP}$, we next present distributions of key variables as a function of local SST, color-coded for each of the five SST quintiles. The left panels of Figure 5 represent the distributions of local SSTs, such that the total of all bins for each TWP SST quintile add up to one. Figure 5a underscores what Figure 2 suggests – the SST mode of about 29.4 °C–29.5 °C is approximately the same regardless of mean SST$_{TWP}$, but the area of the SST mode increases as SST$_{TWP}$ is warmer, and low SSTs cover a smaller area (e.g., red curve versus dark blue curve). The kurtosis of SSTs hence increases with mean TWP SST, without a change in the mode. This is in contrast to when only the northern hemisphere (NH) mean SST is indexed (Figure 5c) or the southern hemisphere (SH) SST is indexed (Figure 5e)—in those subregions, the SST mode tends to increase as the mean SST in either the NH or SH increases.

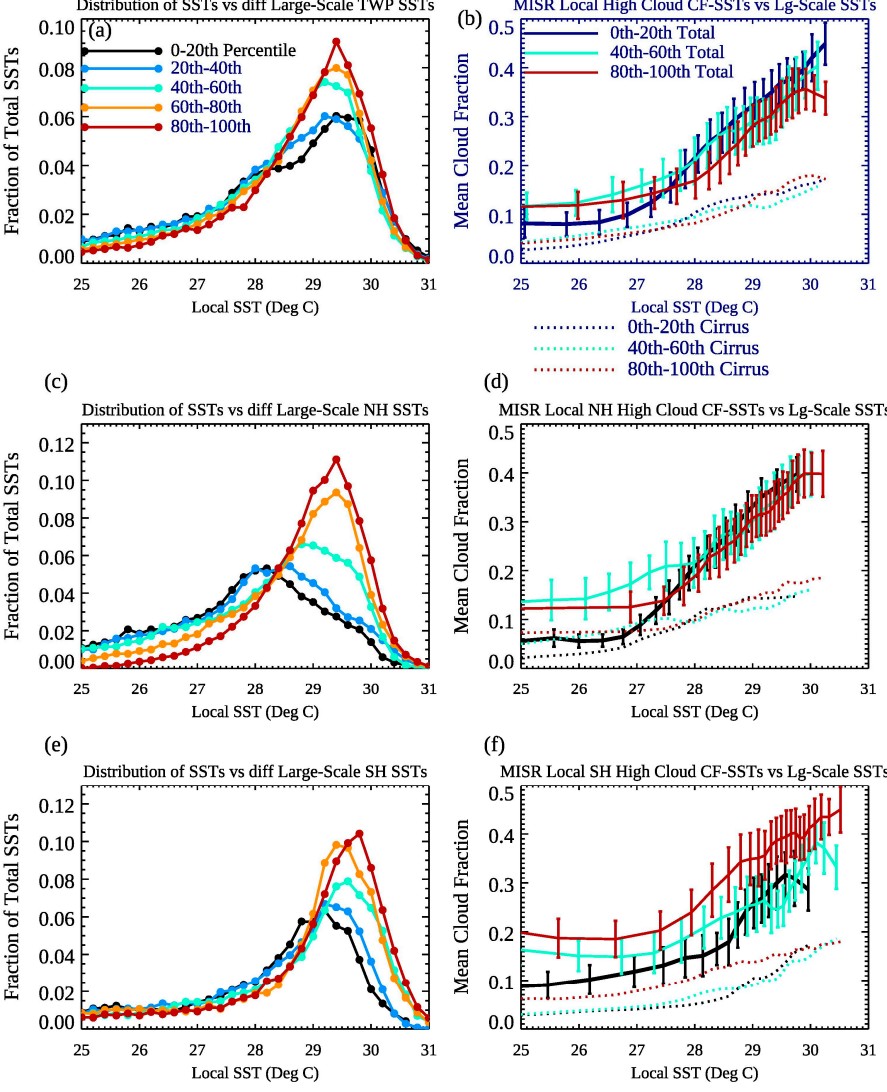

**Figure 5.** (**a**) Distribution of local SSTs over the TWP for raining grids as a function of the TWP SST quintiles in Figure 2 and as described in the text. (**b**) MISR total high cloud fraction versus local SST for raining grids as a function of the first, third, and fifth TWP SST quintiles, along with standard error bars showing the 90% confidence intervals. Dashed lines are for cirrus cloud fraction only. (**c**) Same as (**a**), except for SSTs as a function of SST quintiles in the northern hemisphere only. (**d**) Same as (**b**), except for clouds in the northern hemisphere TWP portion only. (**e**) and (**f**) are the same as (**c**) and (**d**), respectively, except for the southern hemispheric portion of the TWP.

MISR total high cloud fraction versus local SST relationships are shown on the right-side of Figure 5, with separate curves (dashed) for the cirrus-only CF. For the entire TWP, high cloud fraction increases with local SST regardless of SST quintile, though for a given large SST (> 27 °C) there tends to be somewhat more high cloud when the mean TWP SST is cooler. However, the difference is only statistically significant between the lowest and highest TWP SST quintile when local SSTs ≥ 30 °C. This may suggest a local Iris effect, though cirrus cloud versus local SST is quite similar regardless of SST quintile (dashed lines), with a proportionally higher thin cloud fraction of high clouds for higher quintiles for high local SSTs.

In the NH (Figure 5d), except over relatively cool local SSTs in which either the warmest or middle TWP SST quintile has more high clouds over a given SST, the increase of high cloud CF versus local SST is very similar regardless of mean TWP SST. In the SH (Figure 5f), as the subdomain-mean SST increases, the local high cloud fraction is larger for a given local SST for most SSTs, counter to what an Iris Effect would predict. Indeed, there are also more cirrus clouds for a local SST with increasing SST quintile over the SH.

The rest of the analysis of this study deals with only the entire TWP, since only the distribution of SSTs changes rather than the mode as in the NH and SH subregions. In the subsequent figures, we examine how clouds or large-scale dynamics change with SST quintiles partitioned by both local SST and rain; hence, these apply to raining clouds only and are conditional analyses. In Figure 6, we examine anvil plus cirrus clouds (hence all high clouds with $\alpha_{cloud} < 0.6$), and (a) through (e) show how these high clouds vary for the first, second, third, fourth, and fifth TWP SST quintiles. Regardless of mean TWP SST, anvil plus cirrus CF is highest when both local SST is highest—e.g., close to ~30 °C, which is consistent with Kubar et al. [8], and when local rain rate is highest. These and subsequent plots are constructed using 25 bins of both local SST and rain rates within each domain-wide SST quintile, for a combination of up to 625 local SST and precipitation states. There is generally a trend towards an increase of anvil+cirrus cloud fraction extending over lower SSTs from panel (a) to panels (c), (d), and (e) as the mean $SST_{TWP}$ increases, which is quantified in (f). Though there are more anvil+cirrus clouds for the lowest SST quintile over the highest local SSTs ~ 30 °C, there are nearly double the anvil+cirrus clouds over the highest SST quintile for local SSTs < 27 °C. Note also that the SST corresponding to the maximum increase of CF (in either (a) through (e)), or else the SST at which anvil+cirrus begins increasing more strongly with SST, can be considered the convective onset SST, which tends to be about a degree lower (~27 °C) for the lowest versus highest TWP SST quintile. In general, as $SST_{TWP}$ warms, anvil+cirrus clouds are slightly less abundant near the equator, but spread out and linger more meridionally over lower SSTs away from the strongest convection.

In Figure 7, each of the five contour panels from (a)–(e) span the five TWP SST quintiles, but now TOA Net Cloud Forcing is shown from CERES. The trend of TOA Net Cloud Forcing is quite clear—as the mean $SST_{TWP}$ increases, the TOA forcing over the rainiest, most intense convection between ~29–30 °C becomes substantially less negative. In other words, over high local SSTs, the net cooling effects of deep convection reduce as the entire domain warms. We will explore height-albedo (Z-$\alpha_{cloud}$) histograms of cloud fraction and TOA cloud forcing later to better relate the vertical structure of clouds and their reflectivity for lightly, moderately, and heavily precipitating clouds. For now, this sharp increase of TOA net forcing over the highest local SSTs when the domain is warmer in general is a positive regional feedback that warrants attention.

Figure 8 provides an analogous analysis to Figures 6 and 7, now for $\omega_{500}$. When precipitation and local SST are highest, upward motion is stronger when mean TWP SSTs are lower, with a stronger increase of ascent with precipitation. The opposite is true over lower local SSTs—for the highest three SST quintiles, upward motion is stronger averaged over most rain rates over lower local SSTs (especially between 20–25 °C). However, because of significant variability of $\omega_{500}$ for a given local SST, the differences averaged over all rain rates between the highest and lowest SST quintiles are not statistically significant at the 90% confidence level. The average of $\omega_{500}$ over all rain rates as a function of SST is shown for the different TWP SST quintiles in Figure 8f. There is a broad area of stronger

ascent away from the equator when TWP SSTs are warmer, but the lowest TWP SSTs are associated with stronger, concentrated ascent when local SSTs are very high, e.g., SSTs > 29.5 °C, in which ascent at 500 hPa is statistically significantly stronger for the lowest TWP SST quintile versus the warmest TWP SST quintile. This may explain the greater production of high cloud fraction locally over the highest local SSTs, when TWP SST is lower.

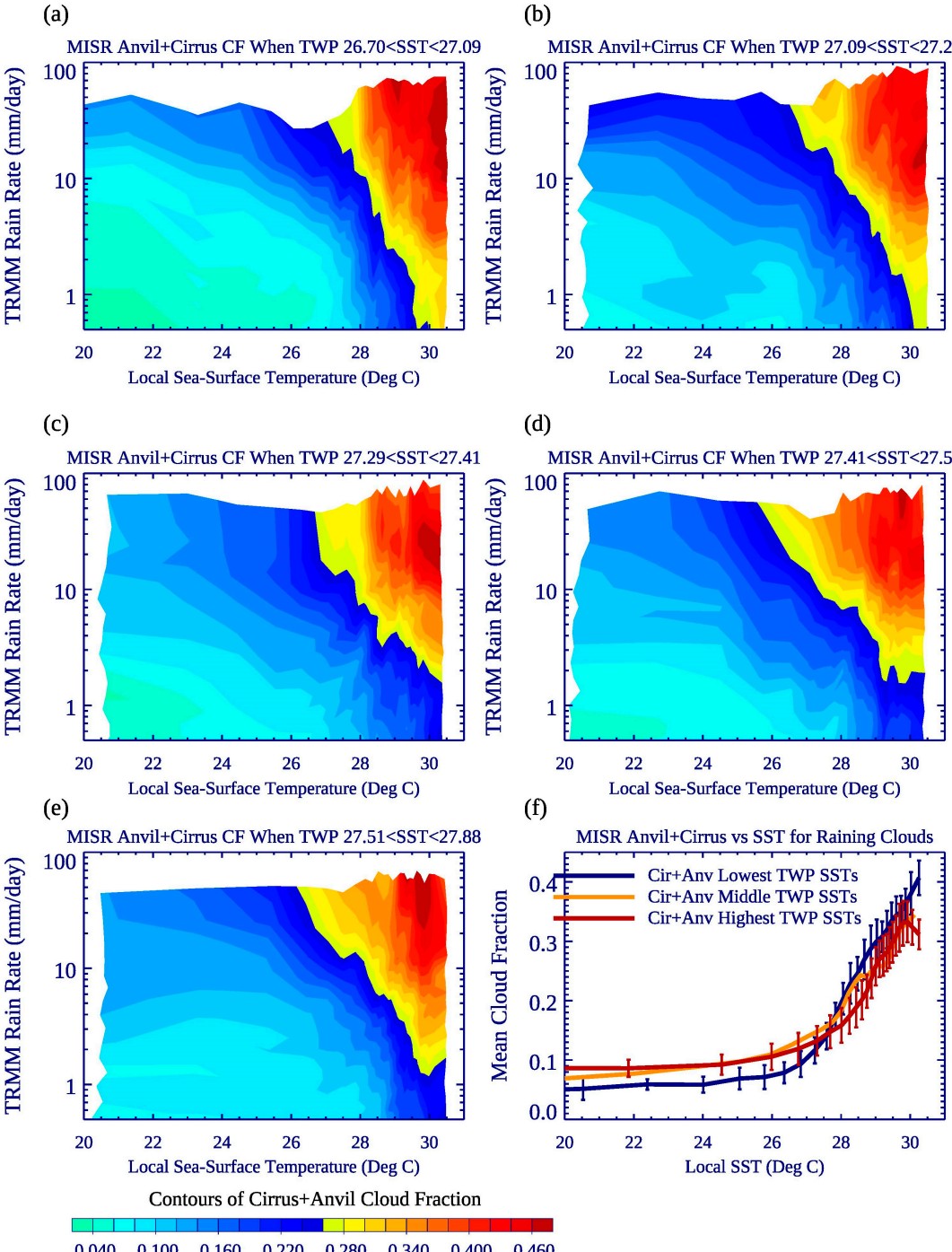

**Figure 6.** (**a**–**e**): Contours of anvil+cirrus cloud fraction as a function of local TRMM rain rate and local SST for each of the five TWP SST quintiles. Each panel has 25 rain rate by 25 SST categories. (**f**) Cirrus+anvil CF versus local SST for the lowest, middle, and highest TWP SST quintiles, averaged over all rain rates. Thick dashed lines depict cirrus CF only. In (**f**), standard error bars for the highest and lowest TWP SST quintiles represent the 90% confidence interval.

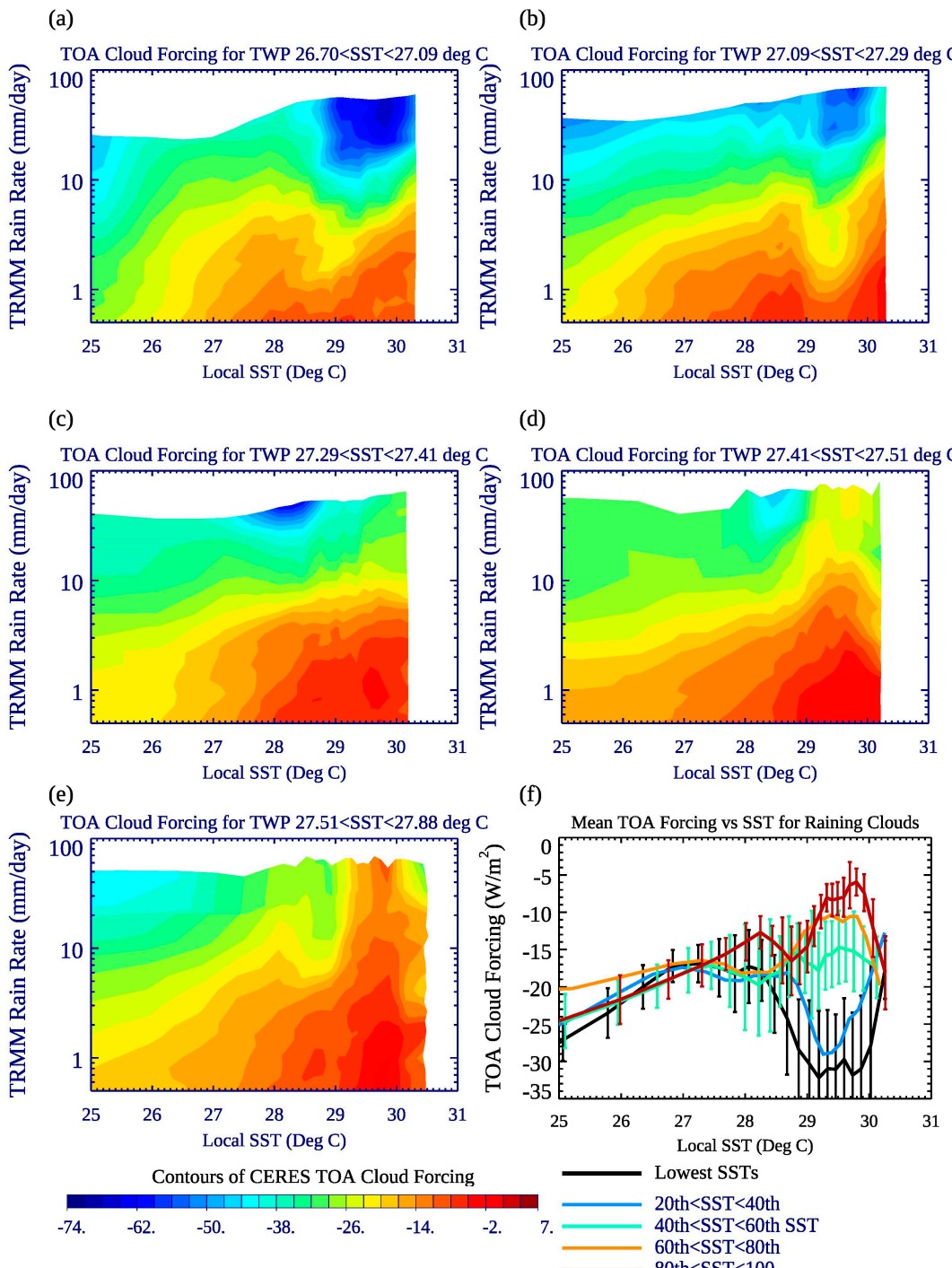

**Figure 7.** (**a**–**e**): Contours of TOA cloud forcing as a function of local TRMM rain rate and local SST for each of the five TWP SST quintiles, for (**a**) first mean TWP SST quintile, (**b**) second TWP SST quintile, (**c**) third TWP SST quintile, (**d**) fourth TWP SST quintile, and (**e**) fifth TWP SST quintile. (**f**) TOA Forcing versus local SST averaged over all rain rates, with standard error bars showing the 90% confidence interval for the lowest, middle, and highest TWP SST quintiles.

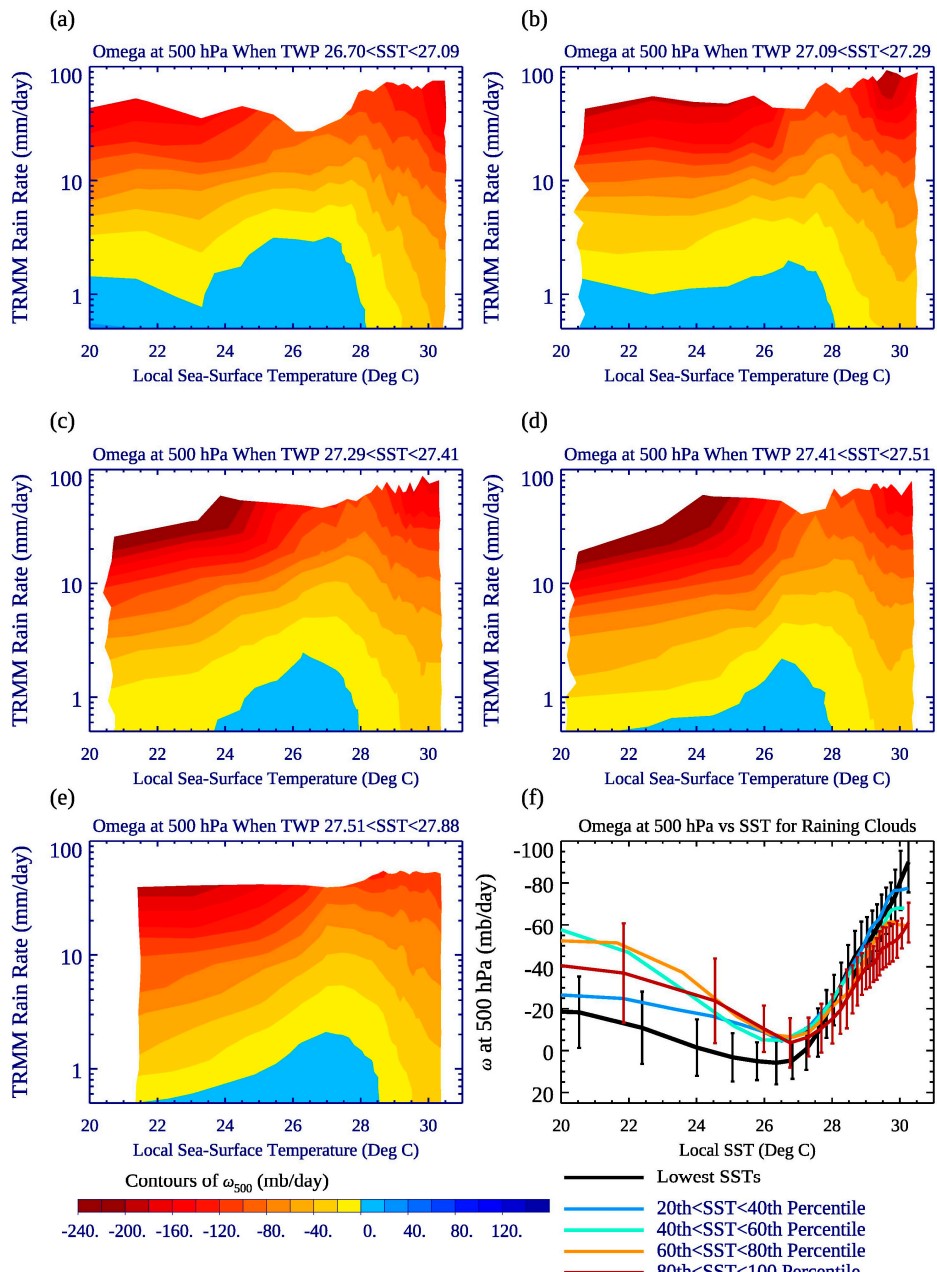

**Figure 8.** (**a**–**e**): Contours of $\omega_{500}$ (mb/day) from ERA-Interim as a function of local TRMM rain rate and local SST for each of the five TWP SST quintiles, for (**a**) first mean TWP SST quintile, (**b**) second TWP SST quintile, (**c**) third TWP SST quintile, (**d**) fourth TWP SST quintile, and (**e**) fifth TWP SST quintile. Results for local SSTs above 20 °C are shown. In panel 8f, standard error bars represent the 90% confidence interval for the lowest and highest TWP SST quintiles for each local SST.

### 4.3. Local Cloud-Rain Rate Relationships for TWP SST Quintiles

As in Kubar et al. [2], we posit that rain rate is not only a good proxy for convective strength, but also that compositing different cloud types as a function of convective strength reveals any possible differences in the nature of horizontal structure and reflectivity structure of convection versus large-scale TWP SST. Figure 9 presents conditional cloud properties or $DIV_{925}$ versus either local rain rate or local SST; the cloud properties are not distributions, nor are they normalized. High cloud properties in (**a**–**c**) are analogous to Figure 6 in Kubar et al. [2], except in that study the West, Central, and East Pacific in the northern hemisphere (boxes between 5–15°N) are analyzed with Aqua MODIS cloud data and AMSR-E rain rates. Here the results are from MISR and TRMM for the larger TWP

domain during different mean SST states. Comparing relationships for different total TWP quintiles of SST effectively reveals insight about the importance of large-scale, non-local mean SSTs.

For rain rates lighter than about 10 mm/day, thick cloud increases similarly with rain rate regardless of TWP SST quintile (Figure 9a). For a given rain rate > 10 mm day$^{-1}$, thick high clouds are marginally more abundant (though not statistically significantly so) for the second lowest or middle TWP SST quintiles, but statistically significantly so at the 90% confidence interval for those two SST quintiles when rain rates exceed 60 mm day$^{-1}$. Anvil cloud fraction is the same for all SST quintiles as a function of rain rate, suggesting that MISR anvil cloud is an excellent predictor of convective strength. We note that there may be some differences between anvil clouds here, which as described earlier are defined by albedos, as compared to anvil clouds in Kubar et al. [2], which are defined by visible τ.

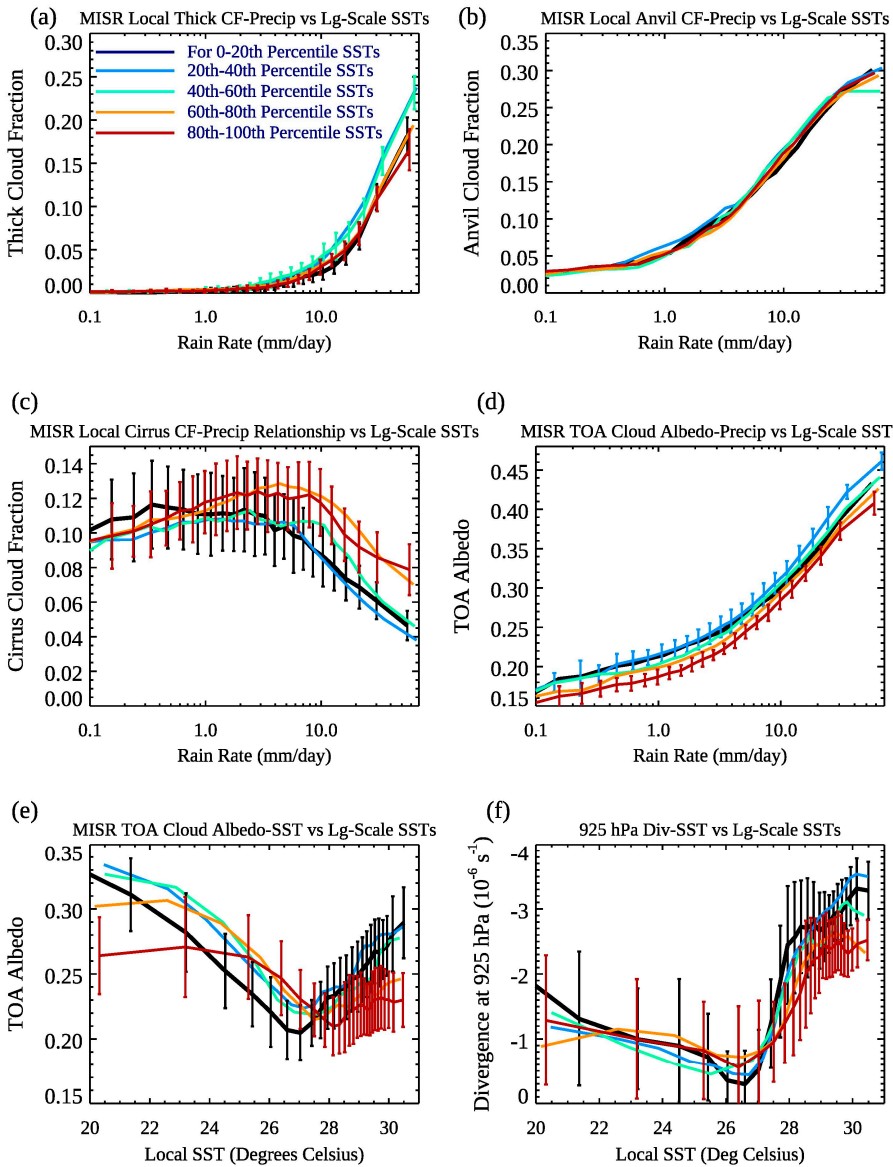

**Figure 9.** (**a**) MISR local thick CF vs. local rain rate for all five TWP SST quintiles. (**b**) Same as (**a**), except for anvil CF. (**c**) Same as (**a**) and (**b**), except for cirrus CF. (**d**) MISR TOA albedo versus local rain rate. (**e**) MISR TOA albedo versus local SST. (**f**) 925 hPa Divergence ($\times 10^{-6}$ s$^{-1}$) vs. local SST. Note that the rain rate axis on (**a–d**) is a logarithmic scale. 90% confidence intervals are shown in panels (**a,c–f**).

For a given rain rate, local cirrus CF is larger when the domain-mean SST is highest, particularly for rain rates > 1 mm/day, with a cirrus CF about 0.03–0.04 higher when the domain-mean SST is

between the 60th–100th percentiles versus the 0th–40th percentiles. However, the differences are statistically significant at the 90% confidence level for rain rates greater than 10 mm day$^{-1}$. This is broadly consistent with Kubar et al. (2007), though the West Pacific cirrus fraction was nearly double the East Pacific cirrus fraction for a given rain rate. Nonetheless, the results here suggest that as the mean TWP warms, deep convection associated with moderate or heavy rain rates has a greater coverage of cirrus clouds. The second lowest TWP SST quintile has the highest TOA cloud albedos for a given rain rate, and compared to the highest TWP SST quintile, the differences are statistically significant at the 90% confidence level (Figure 9d). This is potentially an important positive feedback over the large TWP, for as the TWP SST reaches nearly its warmest levels (second or highest TWP SST quintiles), the shortwave cooling effect of raining clouds is weaker in strength.

Figure 9 concludes with TOA cloud albedo (Figure 9e) and divergence at 925 hPa (Figure 9f) as a function of local SST. As the entire domain warms, the TOA cloud albedo is generally smaller, though this is particularly noticeable over low local SSTs (~20 °C) or local SSTs > 29 °C. Regardless of TWP mean SST, the TOA cloud albedo minimum, around 27–28 °C, represents the transition from shallower to deeper convection. Figure 9f shows weaker convergence at 925 hPa over the highest local SSTs when the domain is warm, in a statistically significant manner at the 90% confidence level for local SSTs ~ 30 °C. This suggests less low-level and mid-level convection as SST$_{TWP}$ warms, and even a possible explanation for reduced thick cloud over the highest SSTs when the domain is warmer. We examine the cloud vertical structure much more in the next section.

### 4.4. *Z-α Histograms of Cloud Fraction and Net Cloud Forcing*

So far, we have primarily focused on cloud amount of different cloud types as a function of local SST and precipitation for quintiles of domain-scale SST, but now we include the vertical axis to quantify how composite clouds change with local precipitation and domain-scale SST. To do this, cloud height–cloud albedo histograms, referred to henceforth as Z-$\alpha_{cloud}$ histograms, are constructed for all raining domains for each of the five TWP SST quintiles. We define light, moderate, and heavy rain categories, with light rain representing rain rates for each of the SST quintiles between the 0.1th and 32nd rain rate percentiles, moderate rain between the 32nd and 68th percentiles, and heavy rain from 68th to 99.9th percentiles. Thus, we have 15 different categories—three local rain categories for each of the five large-scale SST categories. Note that the rain rate categories are calculated from the percentiles within each domain-scale SST; the domain-scale precipitation rate (non-local) is not considered. Again, this figure shows us how the large-scale, non-local SST influences the cloud characteristics of raining convective systems.

We note that the construction of these histograms is motivated by the cloud top temperature/optical depth (T-τ) histograms of Kubar et al. [2]; the histograms presented here are analogs—going up on the *y*-axis means deeper clouds, and going to the right means more reflective clouds. The dimensions of each Z-$\alpha_{cloud}$ bin are 41 (height bins) by 14 cloud albedo bins, with the $\alpha_{cloud}$-bins chosen based roughly on the logarithmic τ bins from Kubar et al. [2], with the following α-midpoints: 0.0346, 0.104, 0.173, 0.242, 0.313, 0.380, 0.45, 0.519, 0.588, 0.657, 0.726, 0.80, 0.873, and 0.958. In Figures 10 and 11, the left column represents lightly raining clouds, the middle column moderately raining clouds, and the right column heavily raining clouds. The domain-mean TWP SST increases from top to bottom. Some general characteristics are as follows:

(1) Regardless of domain-mean-SST, as precipitation increases, high cloud fraction increases, with a shift towards brighter, more reflective clouds, with thick high clouds being more abundant during heavily raining systems. The highest TWP SST quintile also has slightly more thick anvil cloud than lower TWP SSTs for heavily raining systems. Regardless of SST, these clouds are very deep, with tops between 14–15 km.

(2) With increasing rain rate, there is an increase, regardless of SST, of mid-level presumably congestus clouds, centered between 6–7 km. These are moderately reflective clouds, with a mean

albedo of approximately 0.4. Thicker low clouds become less abundant with increasing precipitation category.

(3) The most significant change from cool domain-mean SST to warm domain-mean SST is a decrease in mid-level clouds and a stronger decrease in low-level clouds, regardless of precipitation category. The results for the different cloud categories as a function of domain-mean SST and precipitation category are summarized in Table 1.

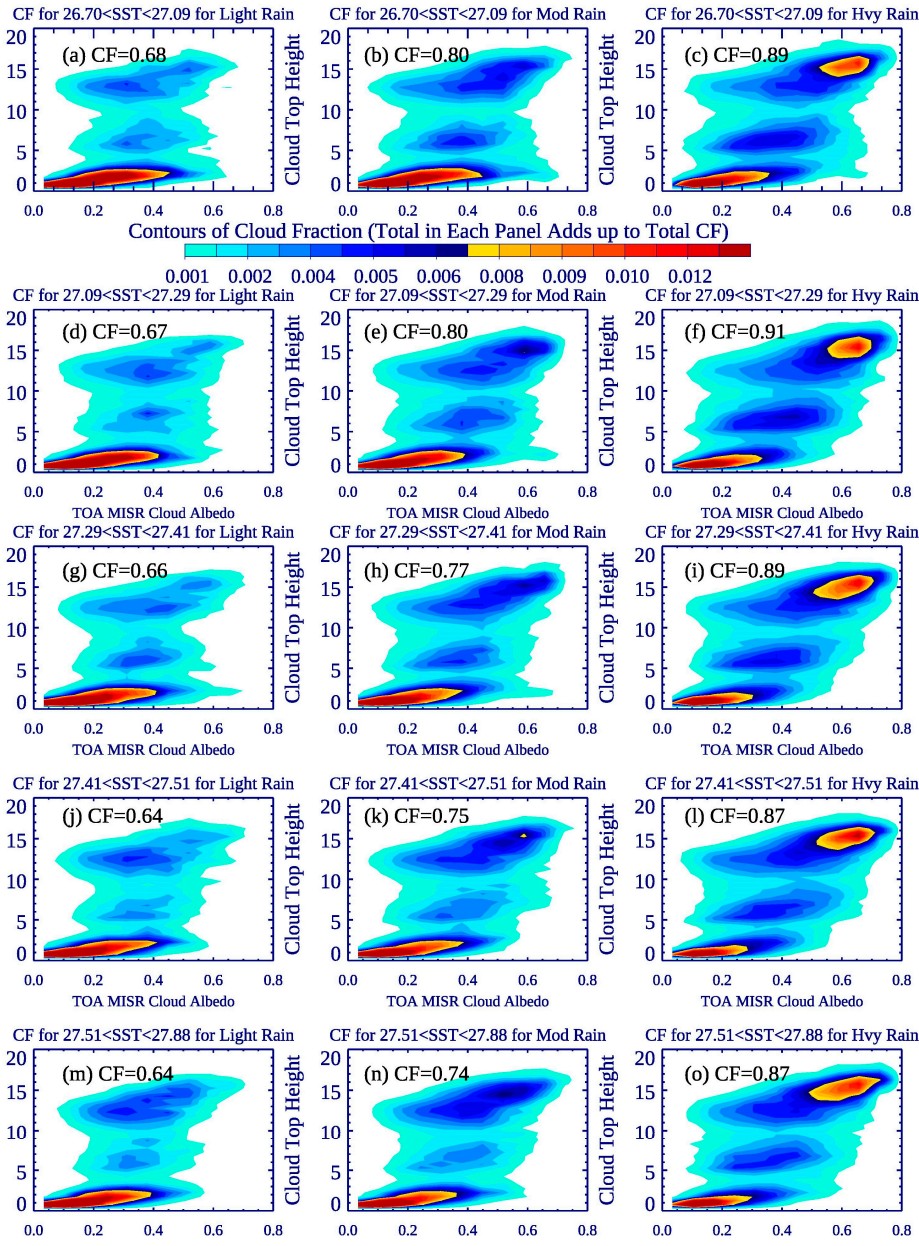

**Figure 10.** Z-albedo histograms of MISR CF for all five TWP SST quintiles for light, moderate, and heavy rain cloud systems. (**a–c**) Histograms when the mean TWP SST is in the first quintile (26.70 °C < SST$_{TWP}$ < 27.09 °C) for (**a**) light, (**b**) moderate, and (**c**) heavy rain cloud systems. (**d–f**) Same as (**a–c**), except when mean TWP SST is in the second quintile (27.09 °C < SST$_{TWP}$ < 27.29 °C). (**g–i**) Same as (**d–f**), except when the mean TWP SST is in the third quintile (27.29 °C < SST$_{TWP}$ < 27.41 °C). (**j–l**) Same as (**g–i**), except when the mean TWP SST is in the fourth quintile (27.41°C < SST$_{TWP}$ < 27.51 °C). (**m–o**): Same as (**g–i**), except when the mean TWP SST is in the fifth quintile (27.51 °C < SST$_{TWP}$ < 27.88 °C). Total cloud fraction is given in each panel.

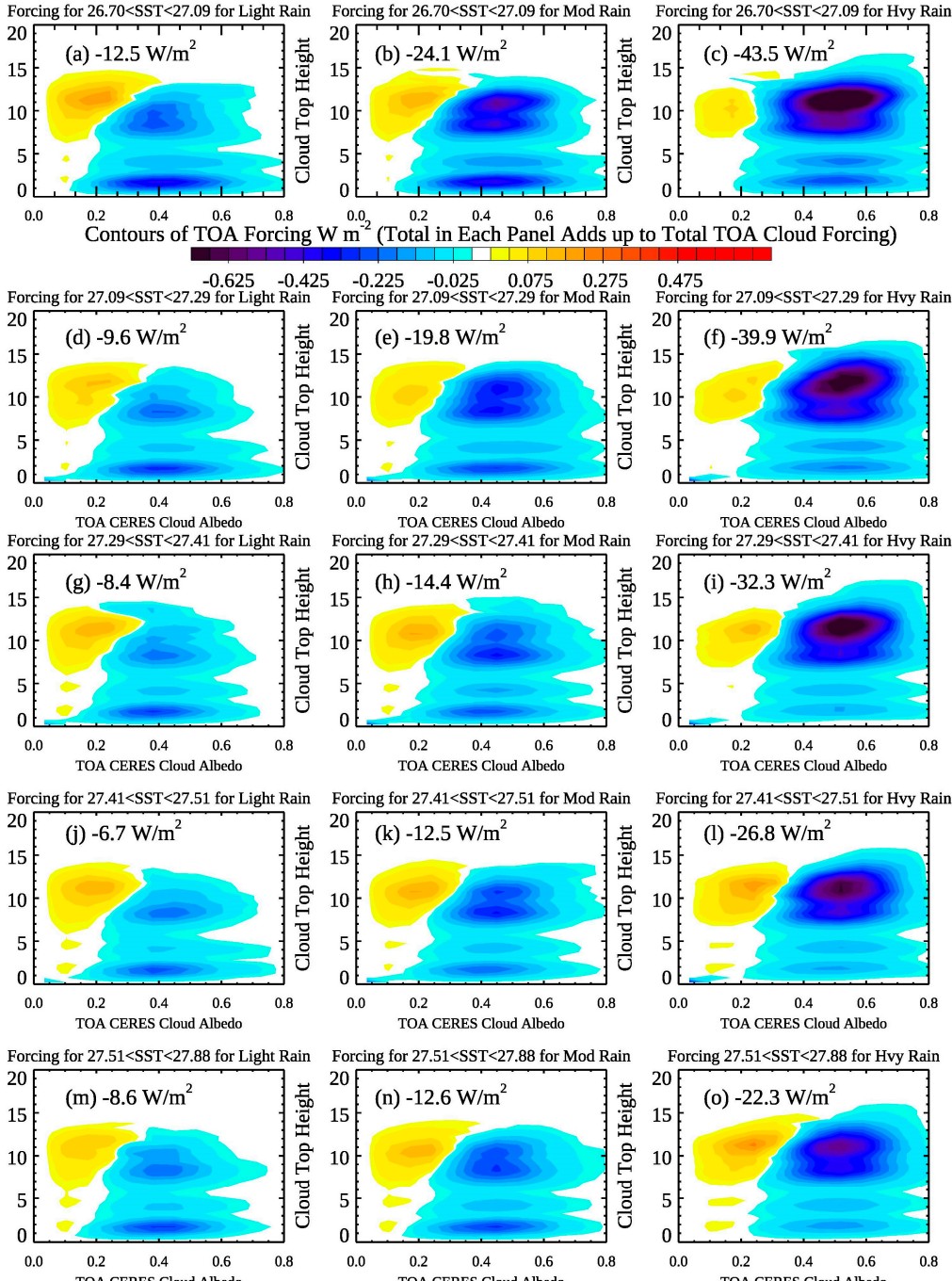

**Figure 11.** Z-albedo histograms of TOA net cloud forcing (W m$^{-2}$) for all five TWP SST quintiles for light, moderate, and heavy rain cloud systems. (**a**–**c**) Histograms when the mean TWP SST is in the first quintile (26.70 °C < SST$_{TWP}$ < 27.09 °C) for (**a**) light, (**b**) moderate, and (**c**) heavy rain cloud systems. (**d**–**f**) Same as (**a**–**c**), except when mean TWP SST is in the second quintile (27.09 °C < SST$_{TWP}$ < 27.29 °C). (**g**–**i**) Same as (**d**–**f**), except when the mean TWP SST is in the third quintile (27.29 °C < SST$_{TWP}$ < 27.41 °C). (**j**–**l**) Same as (**g**–**i**), except when the mean TWP SST is in the fourth quintile (27.41°C < SST$_{TWP}$ < 27.51 °C). (**m**–**o**): Same as (**g**–**i**), except when the mean TWP SST is in the fifth quintile (27.51 °C < SST$_{TWP}$ < 27.88 °C). $\alpha_{cloud}$ is from CERES, and cloud area fraction and effective heights from CERES SYN1DEG are used to construct each histogram; the total of the partial forcings in each histogram add up to the total net cloud forcing in each precipitation category and large-scale TWP SST quintile, given in each panel.

**Table 1.** Cloud Properties from Figure 10 for high clouds, with tops > 9 km (top) and middle-topped and low-topped clouds (bottom). The red number in each column represents the maximum of each column.

| SST Range of Domain-Mean SST | Precipitation (0.0th–32nd Percentile) | Rain Rate (32nd–68th Percentile) | Rain Rate (68th–99.9th Percentile) |
|---|---|---|---|
| | **Cirrus, Anvil, Thick, (Total High CF)** | **Cirrus, Anvil, Thick, (Total High CF)** | **Cirrus, Anvil, Thick, (Total High CF)** |
| 26.70 °C < SST < 27.09 °C | 0.086, 0.116, 0.005, (0.207) | 0.081, 0.189, 0.025, (0.295) | 0.076, 0.240, 0.0893, (0.405) |
| 27.09 °C < SST < 27.29 °C | 0.077, 0.133, 0.009, (0.219) | 0.078, 0.205, 0.036, (0.319) | 0.076, 0.244, 0.120, (0.440) |
| 27.29 °C < SST < 27.41 °C | 0.076, 0.121, 0.012, (0.209) | 0.082, 0.196, 0.041, (0.320) | 0.084, 0.255, 0.115, (0.454) |
| 27.41 °C < SST < 27.51 °C | 0.089, 0.132, 0.013, (0.234) | 0.089, 0.211, 0.031, (0.331) | 0.096, 0.258, 0.107, (0.461) |
| 27.51 °C < SST < 27.88 °C | 0.096, 0.139, 0.011, (0.246) | 0.094, 0.216, 0.030, (0.340) | 0.093, 0.263, 0.108, (0.464) |
| | **Middle, Low Clouds, (Total Mid + Low CF)** | **Middle, Low Clouds, (Total Mid + Low CF)** | **Middle, Low Clouds, (Total Mid + Low CF)** |
| 26.70 °C < SST < 27.09 °C | 0.119, 0.356, (0.475) | 0.157, 0.346, (0.503) | 0.222, 0.263, (0.485) |
| 27.09 °C < SST < 27.29 °C | 0.118, 0.332, (0.450) | 0.166, 0.310, (0.476) | 0.234, 0.234, (0.468) |
| 27.29 °C < SST < 27.41 °C | 0.119, 0.333, (0.452) | 0.150, 0.296, (0.446) | 0.205, 0.228, (0.433) |
| 27.41 °C < SST < 27.51 °C | 0.105, 0.300, (0.405) | 0.145, 0.271, (0.416) | 0.201, 0.212, (0.413) |
| 27.51 °C < SST < 27.88 °C | 0.100, 0.295, (0.395) | 0.128, 0.272, (0.400) | 0.190, 0.216, (0.406) |

Next, we use the same Z-$\alpha_{cloud}$ bins to construct histograms of net cloud forcing from collocated CERES radiative flux data as function of height and TOA cloud albedo. To do this, we composite the TOA forcing in each 1° × 1° grid for the large-scale SST and local precipitation conditions and then scale by the CERES/MODIS cloud fraction from each 1° × 1° for a given TOA CERES albedo to estimate the contribution from each cloud. Adding up all the partial net cloud forcing values in each histogram gives the total estimated net cloud forcing for the precipitation category and large-scale TWP SST, included in each panel. We note that the heights from the CERES SYN1DEG product represent cloud effective heights, which are from MODIS and geostationary remote sensing data, and these tend to be somewhat lower than the MISR cloud top heights. While the histograms in Figure 11 resemble those from Figure 10, there are some notable differences. Clearly anvil and high, thick clouds contribute the most to negative (cooling) TOA forcing, and this effect is strongest for heavily raining clouds, regardless of TWP SST. Lightly raining clouds have a much weaker TOA cooling effect, mostly because lightly raining anvil clouds are less reflective than moderately or heavily raining anvil clouds. For heavy rain, the warming effect of high cirrus clouds becomes stronger as the domain warms, which contributes to the weaker negative radiative effect of raining cloud systems with SST (Figures 9e and 7f). This is consistent with MISR of more cirrus clouds with heavy precipitation systems over the two highest TWP SST quintiles (Figure 10 and Table 1). Overall, precipitating systems, especially moderately raining and heavily raining systems, have a much less negative forcing when the domain is warmer, due to fewer low and middle clouds, and also higher clouds which have a more muted cooling effect.

Results from Figures 10 and 11 are summarized as profiles in the left column of Figure 12, and as a function of TOA albedo on the right column. The left column shows cloud fraction as a function of height for thick (solid), thick+anvil (thick dashed), and all clouds (thin dashed) for the 0th–20th TWP SST percentiles, 40th < SST < 60th TWP SST percentiles, and 80th < SST < 100th percentiles. As from before, total high cloud increases with rain rates, with a greater portion of particularly mid-level clouds when the TWP domain is cooler. The anvil cloud height mode is also slightly lower (~500 m) for moderately or lightly raining clouds for the highest versus the lowest TWP SST quintile. The mode of thick cloud top heights, however, is around 16 km regardless of SST or rain rate, consistent with these clouds being the most undiluted, and with Chae and Sherwood [33], who used MISR Level-2 cloud height data in a partially overlapping region with ours.

The right column of Figure 12 shows the CERES TOA net cloud forcing versus TOA cloud albedo, with two dominant modes, regardless of TWP SST. These are thin clouds dominated by high cirrus clouds, which exert a positive TOA effect particularly when 0.15 < $\alpha_{cloud}$ < 0.25, and then a strong mode negative TOA effect for 0.4 < $\alpha_{cloud}$ < 0.5. The cloud albedo associated with maximum TOA cooling

shifts from 0.4 for lightly raining clouds (Figure 13b) to 0.5 for heavily raining clouds (Figure 13f). That thicker anvil clouds exert the most cooling is consistent with the T-τ net cloud forcing histograms across the tropical north Pacific in Kubar et al. [2]. When the TWP SST is cooler, the peak of negative forcing is more negative, with the effect most dominant for heavily raining clouds. Cirrus clouds with an albedo of ~0.25 exert the strongest warming effect for heavily raining clouds when the mean TWP SST is highest.

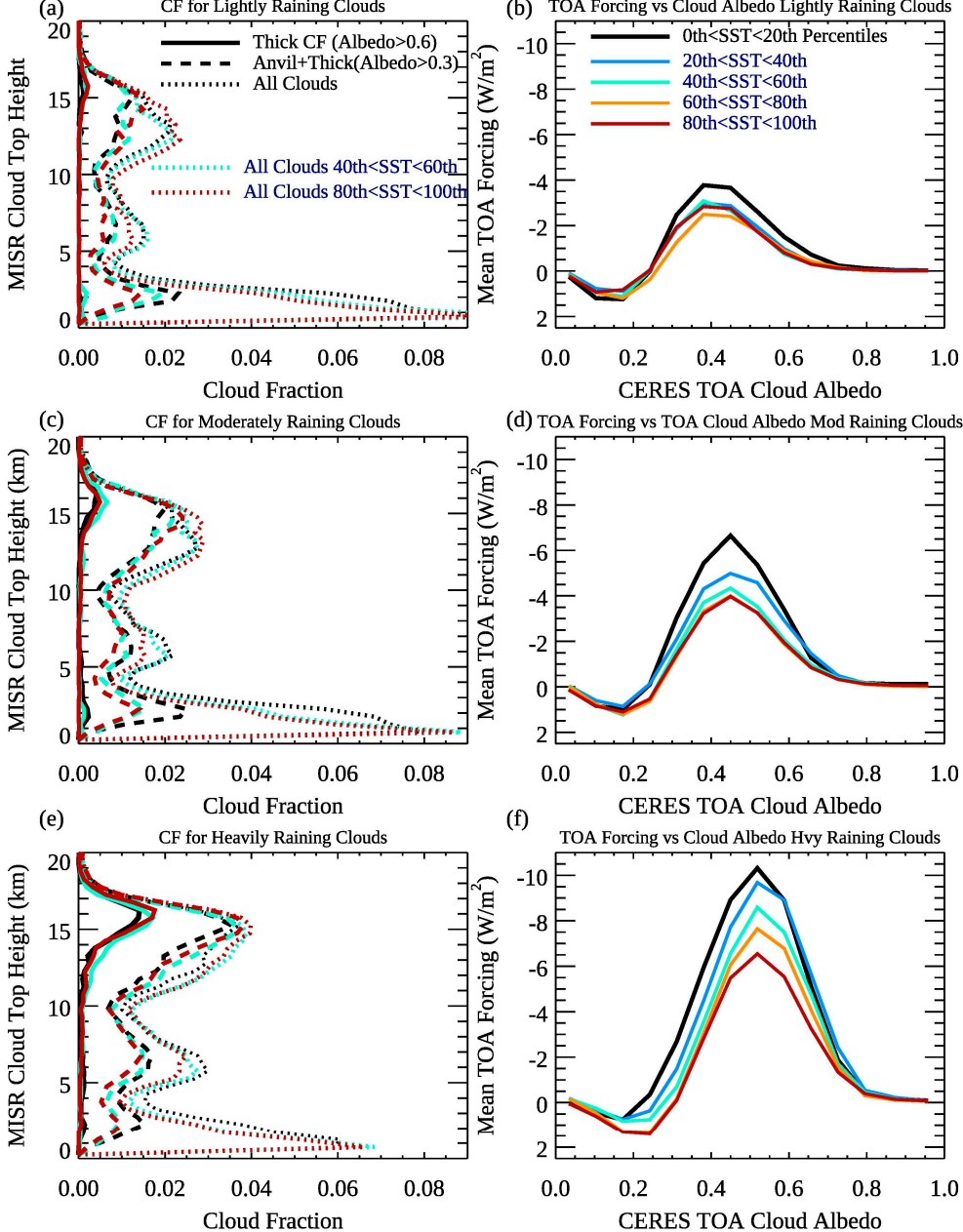

**Figure 12.** (**a**) Cloud fraction profiles for thick (solid), thick+anvil cloud (long dashed), and all clouds (short dashed) for lightly raining clouds for the first, third, and fifth TWP SST quintiles. Here, thick refers to any cloud height with an albedo > 0.6, and thick+anvil any cloud height with an albedo > 0.3. (**b**) CERES TOA cloud forcing for the lightly raining clouds for all five mean TWP SST quintiles. (**c**) Same as (**a**), except for moderately raining clouds. (**d**) Same as (**b**), except for moderately raining clouds. (**e**) Same as (**a**) and (**c**), except for heavily raining clouds. (**f**): Same as (**b**) or (**d**), except for heavily raining clouds.

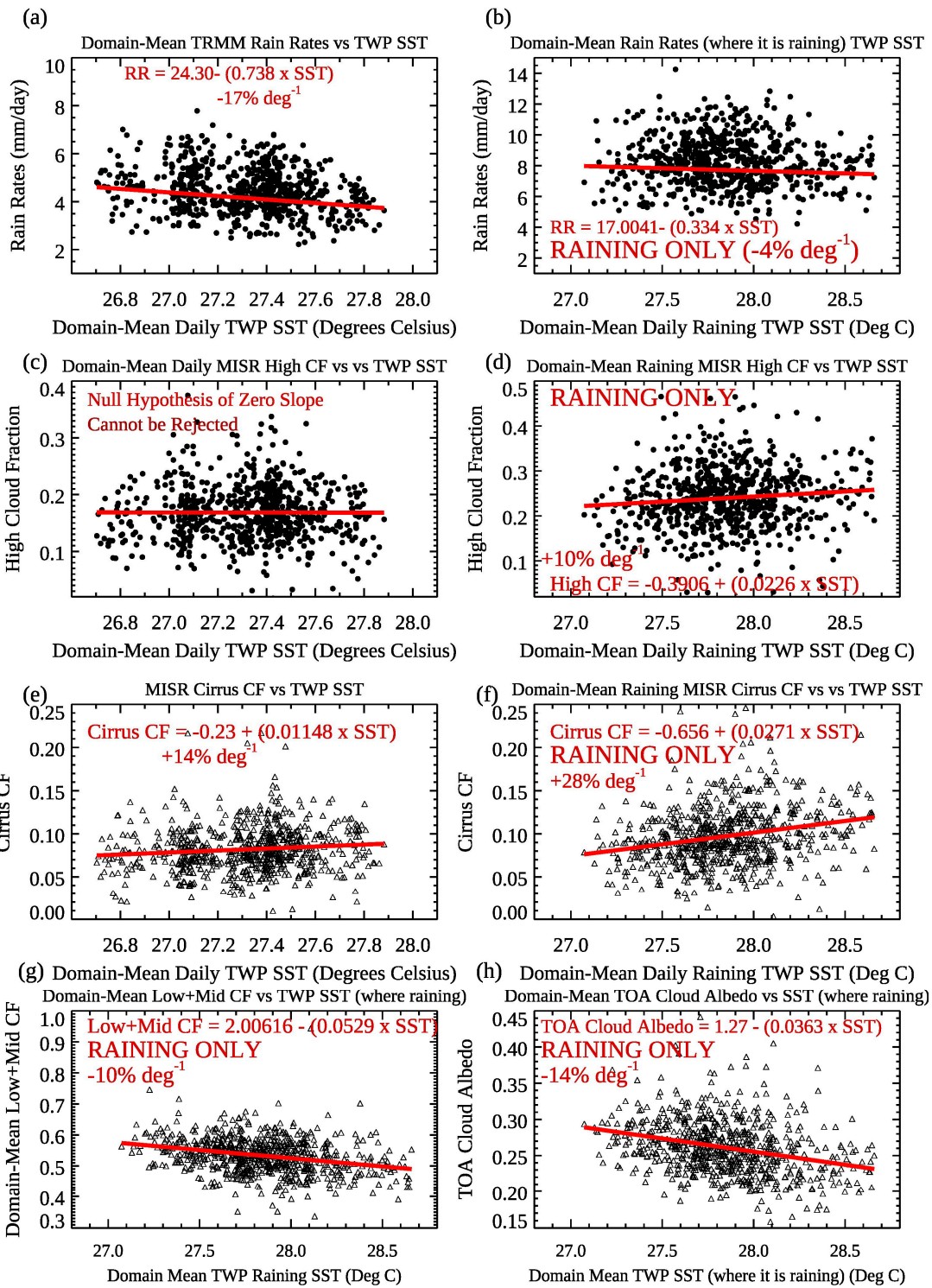

**Figure 13.** Domain-mean properties versus either total domain-mean TWP SST or the raining only TWP SST for (**a**) TRMM rain rates (zero rain rates inclusive) (mm day$^{-1}$), (**b**) TRMM rain rates where it is raining, (**c**) MISR High CF (raining and non-raining), (**d**) MISR High CF where it is raining, (**e**) MISR High Cirrus CF. (**f**) Same as (**e**), except for raining grids only; (**g**) same as (**f**), except for low plus middle-topped clouds; and (**h**) same as (**g**), except for MISR TOA cloud albedo. Linear fit slopes are given in panels where the null hypothesis of a zero slope can be rejected at the 99% confidence interval, and the percent change per degree SST (deg$^{-1}$) is also given in each panel.

### 4.5. Tropical West Pacific Domain Mean Relationships

A good portion of this study so far has focused on cloud and large-scale dynamics relationships normalized by convection rate or local SST, but we wish to take a domain-wide approach of all the variables, similar to how previous studies have addressed large-scale cloud-SST relationships over the tropical Pacific. In Figure 13, we examine domain-mean properties of rain rate and cloud fraction for all grids, for raining-only grids, and then low+middle CF and TOA albedo for raining-only grids for each time step. Panels (a) and (b) show total TRMM rain rates for domain-mean SST and raining-only rain rates versus raining-only domain-mean SSTs, respectively. The total domain scatterplot analysis reveals a decrease of rain rates overall of about $-0.7$ mm day$^{-1}$ deg$^{-1}$ ($-17\%$ deg$^{-1}$). Examining only the raining portion of the domain each day, the slope is smaller at $-0.3$ mm day$^{-1}$ deg$^{-1}$. This result is qualitatively similar to Choi et al. (2017), who broke it down further into convective and stratiform rain rates versus TWP SST. As mentioned in the introduction, that study also used monthly data; ours uses daily data.

When total high cloud fraction for the entire domain is regressed against TWP SST, the slope is close to zero, and a statistical *t*-test cannot reject a null hypothesis of a zero slope. There is, however, a hint of a slight decrease when TWP SSTs are highest. In contrast, when raining-only grids are aggregated, domain-mean high CF increases, with a slope of $+0.0226$ K$^{-1}$ ($+10\%$ deg$^{-1}$), with P(t $\leq -2.71$ or t $\geq 2.71$) $= 0.0092$, which is less than a 0.01 significance level (99% confidence level). When we only examine MISR cirrus clouds, they tend to increase with domain-mean SST, although with a greater slope for raining-only grids of $+0.0271$ K$^{-1}$, or $+28\%$ deg$^{-1}$, and a *t*-value of 6.8. We note that we recently have done a similar analysis with 15-day TWP chunk data using Aqua MODIS, and while total ice clouds tend to decrease slightly with SST, cirrus clouds increase with domain-mean SST, similar to the results presented here. A more quantitative comparison of cloud properties versus TWP SST among different instruments is forthcoming.

Finally, we show domain-mean mid+low level cloud fraction versus TWP SST for raining grids only, with a substantial decrease per degree of SST ($-0.0529$ deg$^{-1}$). This result is qualitatively consistent with the findings from Rapp et al. [29]. This suggests that cloud fraction does indeed aggregate with increasing SST, but not high clouds as has been previously reported, but instead clouds below 9 km. Indeed, the lower-level cloud signal dominates the total CF decrease with TWP SST of $-4\%$ deg$^{-1}$ (not shown). The inclusion of mid-level cloudiness (e.g., clouds down to 500 hPa) could explain part of the decrease of cirrus clouds with TWP SST defined by Choi et al. [32]; if we included middle and high clouds, as a whole they tend to decrease, though with a smaller slope than [32]. The domain-mean TOA albedo of raining grids only versus TWP SST also decreases with a slope of ($-0.036$ deg$^{-1}$ or $-14\%$/deg), suggesting that clouds thin as the entire domain warms, a result we may have predicted from Figure 9e. This is primarily due to an increase in high cirrus clouds and a decrease of low and middle thicker clouds; anvil and thick high clouds, even for raining grids only, are nearly independent of domain-mean SST (not shown).

### 4.6. Robustness and Representativeness of Findings

So far, all of our results have been inclusive of the two-year period from October 2002–September 2004, though as already noted, the first year encompasses a moderate El Niño event, and the second-year represents near-neutral ENSO conditions. In order to better quantify the possible role of ENSO and interannual variability of the tropics, we split the two-year analysis into two individual years, October 2002–September 2003 and October 2003–September 2004. In Table 2, we present a description of principal results from each year, and include the entire two-year period for comparison. This helps highlight results that that are regime (ENSO)-independent and time-independent, and therefore which domain-wide TWP SST-cloud relationships are most robust. We deem results most time-independent those in which the signs of significant slopes are the same for each time period of a given row. Of the relationships explored, slopes of raining-only (SST$_{TWP}$, TRMM rain rates), all or raining only (SST$_{TWP}$, high CF), and all (SST$_{TWP}$, cirrus CF) are time-sensitive, e.g., the top four rows. For instance, during

2002–2003 or the entire period, 2002–2004, TRMM TWP rain rates for raining only grids decreases with SST$_{TWP}$, but during the neutral year (2003–2004), there is no statistically significant slope. Rows two through four of Table 2 suggest that total high cloud or cirrus slopes with SST are less positive during the neutral versus than during a moderate warm ENSO year.

In contrast, the bottom three rows of Table 2 demonstrate statistically significant relationships, regardless of period, of increasing TWP cirrus clouds with TWP SST, and decreasing TOA cloud albedo and low+middle CF with TWP SST for raining-only TWP grids. These three pairs of relationships are thus the most robust domain-mean relationships of our analysis, but including additional years of varying strength and sign of ENSO will be important for future work. Note also that the three robust relationships denote positive SST-cloud feedbacks over the TWP.

**Table 2.** Summary of slopes (from Figure 13) for Oct 2002–Sep 2004, and then each individual year (Oct 2002–Sep 2003 and Oct 2003–Sep 2004). Statistically significant positive slopes are denoted in red, negative slopes in blue.

| TWP Variable Pairs | October 2002–September 2004 | October 2002–September 2003 (Mod. El Nino) | October 2003–September 2004 (Near-Neutral Year) |
|---|---|---|---|
| SST$_{TWP}$, TRMM (Raining Only) | Rain rate decreases by 4% deg$^{-1}$ | Rain rate decreases by 8.5% deg$^{-1}$ | Null Hypothesis of Zero Rain Rate Slope cannot be rejected |
| SST$_{TWP}$, MISR High CF (All) | Null Hypothesis of Zero High CF Slope with SST cannot be rejected | Null Hypothesis of Zero High CF Slope with SST cannot be rejected | High CF Decreases by 14% deg$^{-1}$ |
| SST$_{TWP}$, MISR High CF (Raining Only) | High CF Increases by 10% deg$^{-1}$ | High CF Increases by 15% deg$^{-1}$ | Null Hypothesis of Zero High CF Slope with SST cannot be rejected |
| SST$_{TWP}$, MISR Cirrus (All) | Cirrus CF Increases by 14% deg$^{-1}$ | Cirrus CF Increases by 17% deg$^{-1}$ | Null Hypothesis of Zero Cirrus CF Slope cannot be rejected |
| SST$_{TWP}$, MISR Cirrus (Raining Only) | Cirrus CF Increases by 28% deg$^{-1}$ | Cirrus CF Increases by 32% deg$^{-1}$ | Cirrus CF Increases by 22% deg$^{-1}$ |
| SST$_{TWP}$, Low+Middle CF (Raining Only) | Low+Mid CF Decreases by 10% deg$^{-1}$ | Low+Mid CF Decreases by 11% deg$^{-1}$ | Low+Mid CF Decreases by 8% deg$^{-1}$ |
| SST$_{TWP}$, TOA Cloud Albedo (Raining Only) | Cloud Albedo Decreases by 14% deg$^{-1}$ | Cloud Albedo Decreases by 12% deg$^{-1}$ | Cloud Albedo Decreases by 17% deg$^{-1}$ |

## 5. Conclusions

Using primary satellite data from MISR, with complementary data from CERES, TRMM, and ERA-Interim reanalysis data, this study has focused on how changes of the domain-mean SST of the tropical western Pacific (TWP) influence the redistribution of horizontal SST within the domain and hence local cloud relationships with local SST and rain rate. Going beyond previous investigations which have quantified total TWP high cloud relationships with domain-mean TWP SST, some of which have identified an Iris Effect whereby high cloud amount decreases as the TWP warms, this study has explored sub-TWP relationships of clouds, rain rates, and SST. Furthermore, in this study different cloud types have carefully been differentiated using high-resolution MISR cloud height data in the vertical, with high clouds in each profile defined as those with tops above 9 km, middle clouds tops between 4 km and 9 km, and low-topped clouds below 4 km. Three categories of high clouds, cirrus with cloud albedos < 0.3, anvil clouds with 0.3< albedos <0.6, and thick clouds with albedos > 0.6, closely follow the definition of clouds from a τ-partitioning approach in Kubar et al. [2] and Kubar and Behrangi [8,9]. The partitioning of high clouds based on their albedo in separating clouds most associated with the convective core and those preferentially more abundant at the periphery, or even separated from tropical high cloud systems, is an advantage over previous studies.

The other focus of this study has been how the changes in the TWP SST, which manifest as changes in kurtosis rather than the SST mode, drive changes in near-equatorial SST gradients within the TWP. We have investigated how these in turn drive changes in the local overturning Hadley Circulation within the TWP through the redistribution of low-level divergence (e.g., DIV850) and

mid-level pressure vertical velocity (e.g., $\omega 500$). These circulation changes, separate from any Iris or non-Iris Effect, help explain variability of deep convection and detrained anvil and cirrus clouds close to and well-away from convection. Can the behavior of high clouds as a function of local SST, rain rate, and domain-scale TWP be explained fully by changes in the large-scale dynamics, or are there residual differences as well that must invoke other considerations?

An important finding of this study is that as the mean TWP warms, the north-south warm pool, e.g., the area of SSTs > 27 °C, expands, as does the area of the SST mode. This reduces the north–south temperature gradients, as well as the strength of the low-level convergence and maximum ascent near and just away from the equator. Overall, the overturning circulation weakens as the TWP warms, but moderate ascent extends well away from the equator compared to when the TWP is cooler. This influences the high cloud, rain rate, and local SST relationships as such:

(1) For any domain-mean TWP SST, high cloud fraction of raining clouds increases both with local SST and rain rate, with maximum anvil+cirrus CF maximizing as local SSTs reach 30 °C, and slightly more anvil+cirrus clouds for lower domain-mean TWP periods (e.g., the 0th–20th domain-mean SST quintile versus the 80th–100th domain-mean SST quintile). This coincides with stronger ascent near SSTs of 29–30 °C during lower TWP SST quintiles, but weaker ascent over locally cooler SSTs off the equator compared to mean warm TWP periods.

(2) The net domain-effect changes in the local SST/cloud/precipitation effects is a redistribution of high clouds as a function of domain-mean TWP SST, rather than a net change such that a zero-slope assumption for the regression between TWP SST and TWP high CF cannot be rejected. When only raining portions of the grid are considered, there is an increase of high CF per degree of TWP SST warming of 10% deg$^{-1}$. When only the southern SST is indexed, however, there is a net increase of high cloud amount with SST.

(3) As the TWP warms, the mean net cloud forcing increases (e.g., becomes less negative) by about 10 W m$^{-2}$ per degree of TWP warming for mean raining grids, though locally cloud systems have even more profound differences as a function of mean TWP SST. Where high cloud systems are more prolific, at SSTs around 29°–30°, the TOA net forcing is much less negative for the fifth TWP SST quintile compared the lowest TWP SST quintile, at around -8 W m$^{-2}$ compared to $-30$ W/m$^2$, when averaged over all rain rates. This is due primarily to a greater portion of cirrus clouds, and somewhat less reflective anvil clouds, and fewer moderately thick low and middle clouds, perhaps due in part to the weaker low-level convergence over high SSTs by $0.5 - 1.0 \times 10^{-6}$ s$^{-1}$ (highest versus lowest TWP SST quintiles).

(4) For a given local rain rate, local anvil cloud fraction is the same for all five TWP SST quintiles, making anvil cloud amount a 'universal' proxy for rain rates in the TWP. Since anvil clouds are defined as high clouds with albedos between 0.3 and 0.6, this relationship has ramifications for validation studies of precipitation sensors, as an albedo-based high cloud definition can quantify rain rates over a range of local and large-scale SSTs. In contrast, cirrus clouds are more abundant for a given heavy rain rate (> 10 mm day$^{-1}$) as the entire domain warms, and this characteristic of convection, along with slightly less thick cloud, especially for the most heavily raining clouds, makes deep convective systems less reflective as the domain-mean SST warms.

While total high cloud fraction across the TWP is conservative with respect to domain-mean SST changes, with a near-zero slope as a function of domain-mean SST, high clouds are redistributed, with more anvil + cirrus clouds away from deep convective core clouds, and fewer high clouds over the highest SSTs when the warm pool in the TWP is larger. Total domain-mean cloud cover, including low and middle clouds, is not conservative with TWP SST, as for all grids or raining-only grids, clouds with tops below 9 km decrease, and cloud systems become more top-heavy compared to cooler TWP periods. Thicker lower clouds decrease with TWP SST, suggesting a smaller contribution to total rain when the domain is warmer, consistent with Kubar and Hartmann [3], who show that over the East Pacific ITCZ, which is about 1 °C cooler than the West Pacific, 10% more of the total rain amount falls

from clouds below 9.5 km. The East Pacific may share similar characteristics to the lower TWP SST quintiles in our study.

An advantage of using the entire TWP domain is that the shape of the SST distribution can be used to investigate its role in driving high cloud changes locally and well away from the deepest convective cores. The SST mode is essentially the same for each of the five TWP SST quintiles in this study, but future work could determine if periods exist when the total TWP SST mode changes, including during ENSO events. The stability of the distribution of SST may be key in predicting net changes in total high cloud cover for current and future projected climate warming.

An argument and finding from Lindzen et al. [23] was that detrainment cirrus clouds diminish with increasing SST when normalized by clouds least diluted by entrainment, or driven most by low-level equivalent potential temperature [3]. However, Lindzen et al. [23] used all clouds with brightness temperatures to 260 K to estimate detrainment, which include mid-level clouds not necessarily associated with deep convective clouds. We have shown that for a given heavy rain rate, thick high clouds are more abundant for the second or third lowest TWP SST quintiles (Figure 9a), anvil clouds increase with rain rate similarly for all TWP SST quintiles (Figure 9b), and for a given heavy rain rate, cirrus clouds are more abundant for the two highest TWP SST quintiles (Figure 9c). This would seem to suggest that a greater portion of high clouds linger or become thinner detrainment clouds as the TWP warms, and we briefly examine this by normalizing against thick cloud cover

$$\text{Detrainment Ratio} = [\text{High Cloud Cover - Conv. Core CF}]/\text{Conv. Core CF} \qquad (2)$$

Here, Conv. Core CF is similar to thick high clouds as throughout this study, but with the additional requirement that cloud top heights exceed 15 km. This preferentially targets undiluted, convective core clouds that represent a slightly smaller fraction of thick high CF. Additionally, we perform our analysis against $\omega_{500}$ to attempt to isolate the effect of large-scale SST as best as possible. The joint PDF analysis using the mean values for each $1° \times 1°$ map grid for the first two TWP SST quintiles is shown in Figure 14a, and for the fourth and fifth SST quintiles in Figure 14b. In both panels, the detrainment ratio maximizes for moderately strong upward motion, but then drops off for the strongest ascent, which makes sense as there should be a greater proportion of thick, convective core clouds there.

The other feature is that over the two warmest TWP SST quintiles, for a given vertical velocity, the mode or median of detrainment ratio is larger. This suggests either that there is more direct detrainment from deep convective systems, or that the large-scale environment is more favorable to sustain anvil and cirrus clouds in general for a given amount of convective core clouds. It is important to note that not all of the cirrus + anvil clouds have been sourced to the deep convective core, but the takeaway message is that for a similar composite vertical velocity and thick cloud, cirrus, and anvil clouds do not shrink with large-scale SST, but may slightly expand.

When anvil and cirrus clouds averaged over the entire TWP are studied individually versus domain-mean $\omega_{500}$ and SST (using 100 states of (TWP SST, TWP $\omega_{500}$)), what emerges is two distinct relationships—anvil cloud fraction scales predominantly with domain-mean vertical velocity (Figure 14c), domain-averaged cirrus clouds, scale mostly with TWP SST (Figure 14d). This argues for a positive feedback of cirrus clouds and SST, an argument that has been made before on smaller scales by Kubar et al. 2019 [8,9], but not necessarily in the context of such a large-region as the TWP. While the actual changes of cirrus clouds are fairly small, their response to SST enhances the net warming associated with the loss of low + middle clouds.

We have identified a potentially important positive feedback of a net cloud albedo decrease with TWP SST, even with a modest high cloud increase for raining clouds, and the net effect of this, along with fewer lower-level clouds, yields an increase in net cloud forcing with domain-mean SST. Whether this effect is limited to the TWP, or works more efficiently there, may have implications for helping explain the strengthening Walker Circulation [57], for which the near-equatorial West Pacific has warmed relative to the East Pacific, the latter of which has fewer thinner high clouds. We save a longer-term analysis of the TWP, and of the role of the Walker Circulation, for future work.

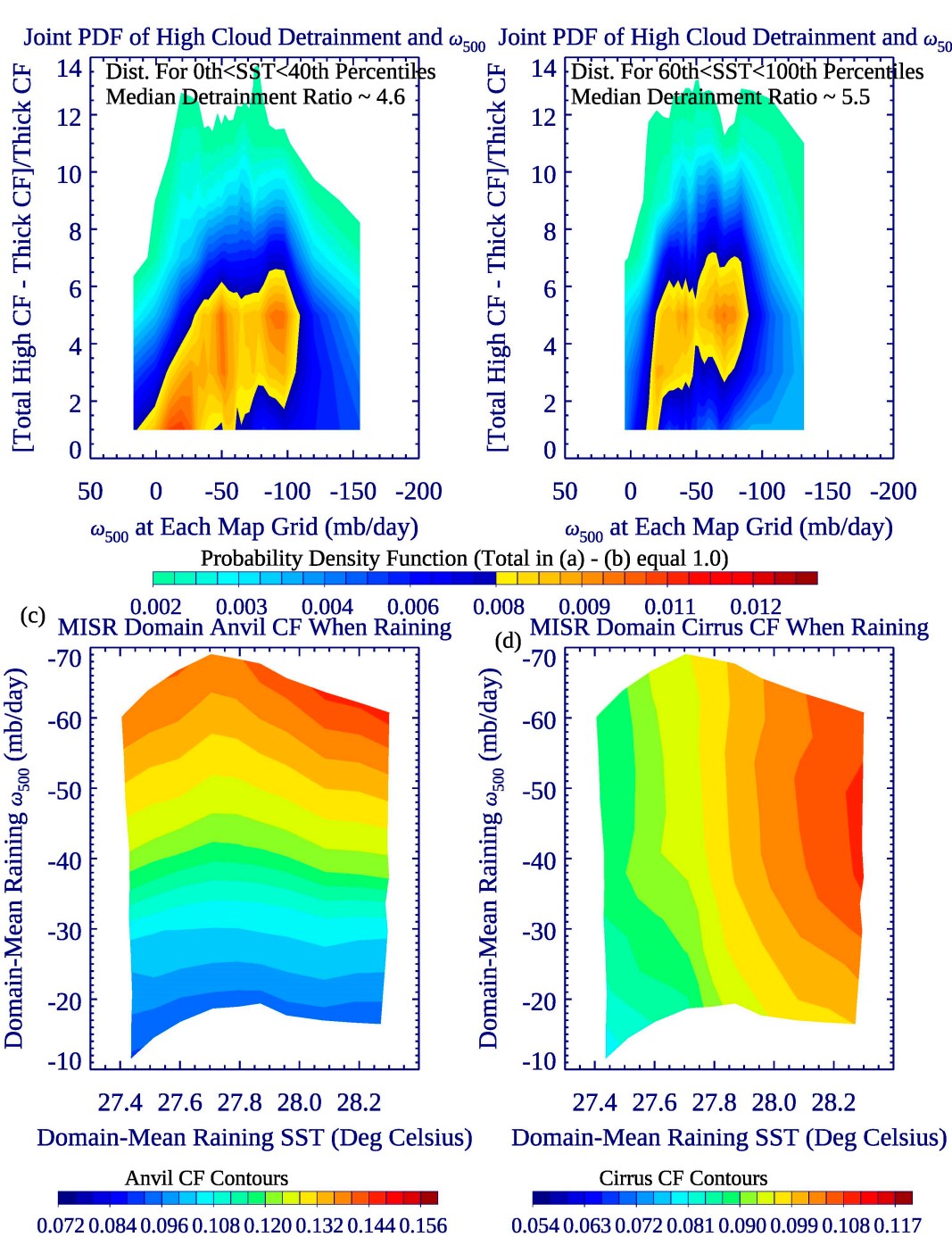

**Figure 14.** (**a**) Joint PDF of high cloud detrainment ratio and $\omega_{500}$ for the first two TWP SST quintiles, collected from the 1° × 1° time-averaged raining map grids during the large-scale SST conditions. Detrainment ratio is estimated from (2) in the text. (**b**) Same as (**a**), except for the upper two TWP SST quintiles. (**c**) Domain-averaged anvil CF versus 100 bins of mean raining (TWP SST, $\omega_{500}$). (**d**) Same as (**c**), except for domain-averaged cirrus CF.

**Author Contributions:** Conceptualization, T.L.K. and J.H.J.; Methodology, T.L.K.; Formal analysis, T.L.K.; Investigation, T.L.K. and J.H.J.; Resources, T.L.K. and J.H.J.; Writing—original draft preparation, T.L.K.; Writing—review and editing, T.L.K. and J.H.J.; Visualization, T.L.K.; Supervision, J.H.J.; Project administration, J.H.J.; Funding acquisition, J.H.J.

**Funding:** This research received no external funding.

**Acknowledgments:** The authors acknowledge the TERRA-MISR Project for support of this project. This research was carried out at the Jet Propulsion Laboratory, California Institute of Technology, under a contract with the National Aeronautics and Space Administration. The authors thank four anonymous reviewers for their helpful feedback and improvement of the manuscript.

**Conflicts of Interest:** The authors declare no conflict of interest.

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
