# Peer review of "Net Cloud Thinning, Low-Level Cloud Diminishment, and Hadley Circulation Weakening of Precipitating Clouds with Tropical West Pacific SST Using MISR and Other Satellite and Reanalysis Data"

_remotesensing, doi:10.3390/rs11101250_

Round 1
Reviewer 1 Report
As I have written in my previous review "The authors combine observational data from many platforms and complement it with ERA reanalysis, in order to derive relations between (mainly high) cloud types and cover, SST regimes, rain rate, and convergence strength. The also provide insight in the existence or not of debated feedbacks. The topic of this manuscript is very important, because the uncertainties in this important geographic area hinder our understanding of the ENSO events, hurt the skill of our climate models, and cast doubt on future climate projections."
General comments for the authors
You have largely addressed my criticisms during the previous round. Along with the improvements suggested by the other reviewers, I think the manuscript improved substantially and therefore I suggest acceptance.
Specific comments for the authors
In the previous round I mentioned random overlap as a possible cause of the opposite changes of low and middle cloud with rain rate. I noticed the opposite changes in the previous version of Table 1 (bottom part). Also you mentioned it yourselves in point (2) of Section 4.4. Your answer on this point was more than satisfactory.
Trivial corrections
l. 283 Please change "assets" to "aspects"(?)
l. 669 Please change "henceforth to" to "henceforth as"
l. 740 "As rain rate increases," could for clarity change to "For heavy rain,"
l. 933 I would delete "per rain rate"
ll. 962-963 Please change Figures 15 with Figures 14
Author Response
Thank you to both reviewers for reading and reviewing our revised manuscript. We are deeply appreciative of your efforts and for the time spent in ensuring our modifications meet the high and strict standards of Remote Sensing, and with these final minor edits suggested, we look forward to advancing our manuscript to the final stage just prior to publication. Through this document, our responses are in “Italics”, below the comments of each reviewer.
Open Review
(x) I would not like to sign my review report
( ) I would like to sign my review report
English language and style
( ) Extensive editing of English language and style required
( ) Moderate English changes required
(x) English language and style are fine/minor spell check required
( ) I don't feel qualified to judge about the English language and style
Thanks again for catching the typos; after perusing our latest version once more, we caught a couple of errors that slipped in there, too, and in concert with the ones below you enumerate, these corrections have helped improved the final version of the manuscript.
Yes | Can be improved | Must be improved | Not applicable | |
Does the introduction provide sufficient background and include all relevant references? | (x) | ( ) | ( ) | ( ) |
Is the research design appropriate? | ( ) | (x) | ( ) | ( ) |
Are the methods adequately described? | (x) | ( ) | ( ) | ( ) |
Are the results clearly presented? | ( ) | (x) | ( ) | ( ) |
Are the conclusions supported by the results? | ( ) | (x) | ( ) | ( ) |
Comments and Suggestions for Authors
As I have written in my previous review "The authors combine observational data from many platforms and complement it with ERA reanalysis, in order to derive relations between (mainly high) cloud types and cover, SST regimes, rain rate, and convergence strength. The also provide insight in the existence or not of debated feedbacks. The topic of this manuscript is very important, because the uncertainties in this important geographic area hinder our understanding of the ENSO events, hurt the skill of our climate models, and cast doubt on future climate projections."
To briefly reiterate our response to the previous review, “Thank you for your thorough and thoughtful feedback, and we appreciate your recognition of the importance of this topic both in our understanding of tropical climate variability, and of cloud-SST feedbacks and interactions over the Tropical Western Pacific (TWP) in observations and as currently simulated/represented in climate models.” Your previous review helped significantly improve the clarity and readability, we feel, of the revised manuscript, and the final comments below provide a final important check prior to what will likely be the final round of the editorial process.
General comments for the authors
You have largely addressed my criticisms during the previous round. Along with the improvements suggested by the other reviewers, I think the manuscript improved substantially and therefore I suggest acceptance.
Thank you again for serving as a reviewer for both versions of our study. Reviewers like you are what help maintain a strong and essential peer-review process, and furthermore enhance the quality of papers whose purpose is to serve and communicate to the scientific (and even non-specialized) community.
Specific comments for the authors
In the previous round I mentioned random overlap as a possible cause of the opposite changes of low and middle cloud with rain rate. I noticed the opposite changes in the previous version of Table 1 (bottom part). Also you mentioned it yourselves in point (2) of Section 4.4. Your answer on this point was more than satisfactory.
Yes, your suggestion of exploring whether or not some of the lowest-level clouds could be seen by MISR in conjunction with higher-topped clouds was important for the purposes of confirming the robustness of the separate and distinct low and middle cloud fraction relationships with rain rate. Thanks again for encouraging us to think further about this, as well as performing a more comprehensive literature review.
Trivial corrections
l. 283 Please change "assets" to "aspects"(?)
Thank you; we agree and have made this change. Originally, we used the word “assets” as we were trying to convey advantages of the MISR configuration, but after further consideration don’t feel this worked well and using “aspects” instead is more appropriate.
l. 669 Please change "henceforth to" to "henceforth as"
Thank you; this was a typo and we have changed to “henceforth as.”
l. 740 "As rain rate increases," could for clarity change to "For heavy rain,"
Thanks for this helpful suggestion, which makes sense in particular since cirrus cloud positive radiative forcing exhibits greater variability among different TWP SSTs only for the heaviest rain category, and is most positive for this category.
l. 933 I would delete "per rain rate"
Thank you; yes, we agree after re-reading this sentence that “per rain rate” was both unnecessary and redundant, and it has been removed.
ll. 962-963 Please change Figures 15 with Figures 14
Thank you for catching this; we have now changed these two instances of Figure 15 (original manuscript reference!) to Figure 14.
Submission Date
17 April 2019
Date of this review
09 May 2019 12:54:30

Reviewer 2 Report
The authors have addressed most of my comments. I recommend the manuscript for publication.
Author Response
Thank you to both reviewers for reading and reviewing our revised manuscript. We are deeply appreciative of your efforts and for the time spent in ensuring our modifications meet the high and strict standards of Remote Sensing, and with these final minor edits suggested, we look forward to advancing our manuscript to the final stage just prior to publication. Through this document, our responses are in “Italics”, below the comments of each reviewer.
Open Review
(x) I would not like to sign my review report
( ) I would like to sign my review report
English language and style
( ) Extensive editing of English language and style required
( ) Moderate English changes required
(x) English language and style are fine/minor spell check required
( ) I don't feel qualified to judge about the English language and style
Yes | Can be improved | Must be improved | Not applicable | |
Does the introduction provide sufficient background and include all relevant references? | (x) | ( ) | ( ) | ( ) |
Is the research design appropriate? | (x) | ( ) | ( ) | ( ) |
Are the methods adequately described? | (x) | ( ) | ( ) | ( ) |
Are the results clearly presented? | (x) | ( ) | ( ) | ( ) |
Are the conclusions supported by the results? | (x) | ( ) | ( ) | ( ) |
Comments and Suggestions for Authors
The authors have addressed most of my comments. I recommend the manuscript for publication.
To reiterate our sentiments from the first round of reviews, thank you for taking the time to thoroughly review our manuscript; we appreciate your willingness to review the revised version as well, and appreciate the encouraging statements. We feel that the review process in general has greatly improved our manuscript, and are looking forward to proceeding to the next step of the publication process.
Submission Date
17 April 2019
Date of this review
25 Apr 2019 08:11:29

This manuscript is a resubmission of an earlier submission. The following is a list of the peer review reports and author responses from that submission.
Round 1
Reviewer 1 Report
Review of “The Importance of the Redistribution of Local SSTs on Cloud Thinning, Transition Cloud Top-Heaviness, and Hadley Circulation Weakening With Tropical West Pacific SST Using MISR, CERES, TRMM and Reanalysis Data”
General comments:
This manuscript mainly investigates the relationship between mean TWP STT and distribution of local STT as well as the impacts of local STT distribution on cloud and rain. Some of descriptions are inconsistent with figures. Additionally, there are many grammatical errors and sentences that are very difficult to understand. This manuscript should be rewritten throughout before consideration of publication.
Specific comments:
1. Line 538-539: “However, for heavier rain rates, there is more thick cloud for a given rain rate when the mean TWP is cooler”. This description is inconsistent with figure 9a. In figure 9a, there is apparently less thick cloud for coolest SST (0-20th percentile STT) than 20-40th percentile STT and 40-60th percentile STT when rain rate is larger than 10.0 mm/day.
2. Line 551-552: “Indeed, the TOA albedo for any given rain rate is smaller as the mean TWP warms (figure 9d)”. This description is in consistent with figure 9d. In figure 9d, higher SST (20-40th percentile and 40-60th percentile) can show larger TOA albedo than lower SST (0-20th percentile).
Reviewer 2 Report
The authors explore cloud fraction variation with local SST and rain rate and how the local relationships are impacted by domain mean SST and horizontal SST structure. The literature survey is adequate, and the method and materials are well described. The authors have supported the conclusions with appropriate analysis. My comments are as follows:
Minor comments:
1. Line 40: Typo; additional full stop.
Major comments:
1. The author’s use only two years of data from October 2002 to September 2004. There are two issues with this. Firstly, Davis et al. (2017) reported artefacts in the data with a shift in the position of sun‐glint that occurred during the first two years which may have affected cloud height. Has this been corrected in the CFbA product? Have the authors tried using later years of the mission data to see any major differences in results due to sampling issues in the early years of the MISR mission? Secondly, there was El Niño of moderate intensity (McPhaden (2003)) developed in the tropical Pacific in 2002/03. How much impact would this have had on the results? Have the authors compared neutral ENSO years?
https://agupubs.onlinelibrary.wiley.com/doi/full/10.1002/2017JD026456
https://journals.ametsoc.org/doi/pdf/10.1175/BAMS-85-5-677
2. The authors use a multi-sensor and multi-dataset approach to investigate relationships, but multi-sensors have different sensitivity, may have slightly different viewing geometry and may not be looking at the same type of clouds due to time differences? How much effect will this have on results, especially at local timescales? CERES, MISR, MODIS, TRMM all have different footprint coverage? There have been studies in the past collocating CERES, MISR and MODIS dataset?
https://agupubs.onlinelibrary.wiley.com/doi/full/10.1029/2006JD007146
3. The authors use ERA-Interim dataset for large-scale dynamics, which is the usual practice since such direct measurement are not available. However, the authors do not compare cloud fraction by altitude product with a cloud fraction (reproduced as altitude products) produced by ERA-Interim dataset. How would they differ? Otherwise, why not just use ERA-interim dataset (including forcing and cloud fraction data) for full analysis for completeness.
4. Evident from Figure 1(a), MISR misses thin cirrus with optical depths < 0.3. How is this mismatch explained as current cloud fractions explored does not include subvisual cirrus, and yet most of the argument is based on the iris effect related to cirrus clouds? Is there any scope for looking at CALIPSO dataset?
Reviewer 3 Report
Being strong modulators of the Earth's energy budget, the clouds in the climate system continue to challenge the prediction of future climate. This paper, with joint data from different satellite instruments and reanalysis, investigates the relationships between local/domain averaged SST and cloud/precipitation in the TWP region. The present research could be important in reconsidering cloud, in particular high cloud, feedbacks in the context of a warming climate.
Overall, this research is topical and the conclusions could be important. However, I do have several concerns about the data and corresponding data analyses:
First, cloud data from two different instruments, namely MISR and CERES, are used. Some properties, for example cloud optical thickness, albedo, TOA flux, are provides by both of them. It is not clear in general why the author want to use one from a specific instrument. For example, is there a reason that you chose cloud optical thickness from CERES but cloud albedo from MISR?
Second, cloud fraction is an important variable in the study. As far as I know, only with passive sensors, it is very difficult to deal with multi-layer clouds. Even if a multi-layer cloud scene is detected with MISR multi-angular observations, let's say anvil (or cirrus) over low cloud, how to extract optical thickness of the uppermost anvil layer? If the multi-layer case is eventually grouped to "high and thick cloud" category, how could this impact the conclusions in this study? Or in other words, how could this impact overall cloud fraction for each type?
Third, I am confused about several definitions through out the paper:
Equation 1: Why did you use cloud albedo from MISR and clear-sky albedo from CERES? Cloud fraction in equation 1 is from MISR?
Section 3: the three cloud categories (low, middle, and high-topped cloud) are defined based on MISR retrieval or CERES effective height?
Distribution curves shown in Figs. 5, 9 and others: how are they normalized?
MISR TOA albedos shown in Fig. 9:not clear to me if they are cloudy-sky albedos or all-sky albedos? In Fig. 12, why the x-label changes from "CERES TOA cloud albedo" in (b) and (d) to "CERES TOA albedo" in (f)? Is all-sky albedo used in panel f?
In my opinion, this research could be important and the paper could be accepted if the authors could make it more clear to understand based on (but not limited to) my comments listed above. So I suggest a "Reconsider after major revision".
Reviewer 4 Report
The authors combine observational data from many platforms and complement it with ERA reanalysis, in order to derive relations between (mainly high) cloud types and cover, SST regimes, rain rate, and convergence strength. The also provide insight in the existence or not of debated feedbacks. The topic of this manuscript is very important, because the uncertainties in this important geographic area hinder our understanding of the ENSO events, hurt the skill of our climate models, and cast doubt on future climate projections.
General comments for the authors
The authors provide an exhaustive analysis of relevant cloud properties. I don't have many objections or comments on the methodological details. The manuscript however could be significantly improved in the presentation and the results communication area. At points I found myself confused by reading unclear, unnecessarily convoluted sentences, on secondary details, mentioned and explored in multiple sections.
Specific comments for the authors
Introduction: At points, too cumbersome with details. As an example, in ll. 53-55 you define the anvil clouds in another paper and compare anvil clouds between WP and EP. EP is not examined in this manuscript, so details such as these could be removed. This is only one example, but there more scattered around.
l. 193: "October 2002 through September 2004". Please mention why this period, containing a medium-strength positive ENSO event, was selected. Also, please comment on the possible differences in the study results, if a different period with either a strong ENSO event or with neutral conditions were being used.
ll. 260-261: "but as the focus is quantifying the radiative effects of both large-cloud systems or the composite of many individual clouds, this is not considered a shortcoming" Unclear. How is this sentence related to the temporal characteristics of MISR and CERES datasets?
l. 325: "for raining grids". The "raining" property of a grid point is first mentioned here, but it applies to many places in the study. I fail to understand why you use the occurrence of rain as a filter for grid points. I understand the need to derive relationships between cloud properties and rain. However, in a study with one goal to determine the effect of SST on high cloud properties and high cloud radiative effects, I do not see why only raining grid points are used over and over again. Could you explain more why you use the raining property to screen grid points?
l. 441: "... that the peak SST ..." You use peak SST to describe the most common SST value or the mode of the histogram, don't you? The use of "peak" is rather confusing, at least to me, because I associate it with the maximum SST.
ll. 607-610: The mid-level and low clouds change in opposite ways with the rain rate. In some studies this has been attributed to possible problems with cloud overlap detection. Could this be an issue for MISR?
Table 1: For the low precipitation column and low TWP-SST case (and maybe others), the sum of middle, low, high cloud CF is 0.12+0.36+0.22=0.68, which is slightly different from the CF in Fig. 10a.
ll. 633-645: "and less because cirrus clouds are more widespread". In contrast, it appears from Table 1 that the cirrus clouds change only marginally and in mixed directions with rain rate. Also in 638-639 you almost make the opposite claim and there too, the change of cirrus with rain rate is marginal. These results do not seem very robust.
Figure 13: Fig 13c is not referenced in the text. Also Figures 13a and b are referenced in lines 666-671. All pieces of information provided in these lines are supported by Table 1. Therefore, Figure 13 is in my opinion unnecessary.
l. 781: The detrainment ratio is mentioned here, but defined in the next page.
Trivial corrections
ll. 32-34: "While more cirrus clouds for heavy precipitating systems exert a stronger positive TOA effect when the TWP is warmer, anvil clouds are also less reflective and have a weaker cooling effect when the TWP is warmer." "When the TWP is warmer" is unnecessarily mentioned twice.
ll. 69-70: "Though the smaller WP within the north Pacific ITCZ has relatively weak meridional SST gradients compared to further east,... " Unnecessary and furthermore already mentioned above.
l. 90: "but in the net may or may not change total cloud cover." Please simplify and remove
ll. 106-107: ", as atmospheric circulation processes are neglected". Unclear, please remove
ll. 110-117: The presentation of the Iris effect is not clear for anyone unfamiliar with it. First, "consolidation" could mean strengthening of the convection and more cirrus, as well as denser spatial organization and less cirrus. Second, the thermal infrared effect of the changing cirrus is not mentioned at all.
ll. 133-136: Unclear. Please explain further if this point is important, or remove.
l. 140: "secondary". Do you mean "second" or "latter"?
ll. 159-160: Please change to "... test how SST distribution properties such as kurtosis change with ..."
l. 163: Please remove "absolute"
l. 219: Please remove "provide"
l. 310: "the latter of which modulates" should be changed to "which also modulates"
ll. 335-336: "... the middle of the mode..." should be just "mode"
l. 346: Please change "show" to "shown"
ll. 401-402: "Then, minimal high clouds reemerge..." In this phrasing it is implied that high clouds were absent but then reemerge, albeit minimally. I would phrase it "Then, high clouds become scarce in ..."
l. 428: "except that there is minimal subsidence" When reading it, I placed emphasis on the existence of subsidence, when I should place emphasis on "minimal". So I would write it as "except that there is only minimal subsidence". However, you could consider removing the middle column of Figure 4 for the 850 hPa ω altogether, since all relevant information is provided better by the last column for the divergence.
l. 437: Please change "the how" to "how the"
ll. 450-451: "for a given SST there tends to be somewhat more high cloud over higher SSTs (e.g. local SSTs > 27°C) when the mean TWP SST is cooler" could be changed to the clearer "for a given large SST (> 27°C) there tends to be somewhat more high cloud when the mean TWP SST is cooler"
ll. 626-627: "from the CERES SYN1DEG product" is written twice in the same sentence
l. 638: An optional suggestion: Due to the extensive information provided in the paper, the reader sometimes is overwhelmed. It would be helpful to include references to the relevant Figures that support some statements. In this case "..., which contributes to the weaker negative radiative effect of raining cloud systems with SST (Figures 9e, 7f)."
l. 638: Please change to "This is consistent with MISR ..."
l. 673: "indeed diminish as TWP SST increases" should probably by changed to "indeed results in diminishing CF with increasing TWP SST" or something like it.
l. 830-832: "We have shown that for a given rain rate, thick high clouds are more abundant over the heaviest local rain rates when the TWP SST is lower..." could be simplified to "We have shown that for a given heavy rain rate, thick high clouds are more abundant when the TWP SST is lower (Figure 9a)"
l. 840: "clouds, a represents" to "clouds, that represent"
l. 861: "in the context of a" to "in the context of"